# High-probability complexity guarantees for nonconvex minimax problems

**Yassine Laguel**[*]
Laboratoire Jean Alexandre Dieudonné
Université Côte d'Azur
Nice, France
`yassine.laguel@univ-cotedazur.fr`

**Yasa Syed**
Department of Statistics
Rutgers University
Piscataway, New Jersey, USA
`yasa.syed@rutgers.edu`

**Necdet Serhat Aybat**
Department of Industrial Engineering
Penn State University
University Park, PA, USA
`nsa10@psu.edu`

**Mert Gürbüzbalaban**
Rutgers Business School
Rutgers University
Piscataway, New Jersey, USA
`mg1366@rutgers.edu`

## Abstract

Stochastic smooth nonconvex minimax problems are prevalent in machine learning, e.g., GAN training, fair classification, and distributionally robust learning. Stochastic gradient descent ascent (GDA)-type methods are popular in practice due to their simplicity and single-loop nature. However, there is a significant gap between the theory and practice regarding high-probability complexity guarantees for these methods on stochastic nonconvex minimax problems. Existing high-probability bounds for GDA-type single-loop methods only apply to convex/concave minimax problems and to particular non-monotone variational inequality problems under some restrictive assumptions. In this work, we address this gap by providing the first high-probability complexity guarantees for nonconvex/PL minimax problems corresponding to a smooth function that satisfies the PL-condition in the dual variable. Specifically, we show that when the stochastic gradients are light-tailed, the smoothed alternating GDA method can compute an $\varepsilon$-stationary point within $\mathcal{O}(\frac{\ell\kappa^2\delta^2}{\varepsilon^4} + \frac{\kappa}{\varepsilon^2}(\ell + \delta^2\log(1/\bar{q})))$ stochastic gradient calls with probability at least $1 - \bar{q}$ for any $\bar{q} \in (0, 1)$, where $\mu$ is the PL constant, $\ell$ is the Lipschitz constant of the gradient, $\kappa = \ell/\mu$ is the condition number, and $\delta^2$ denotes a bound on the variance of stochastic gradients. We also present numerical results on a nonconvex/PL problem with synthetic data and on distributionally robust optimization problems with real data, illustrating our theoretical findings.

## 1 Introduction

Minimax optimization problems arise frequently in machine learning (ML) applications; indeed, constrained optimization problems such as deep learning with model constraints [24], dictionary learning [10, 57] or matrix completion [28] can be recast as a minimax optimization problem through Lagrangian duality. Other applications include but are not limited to the training of GANs [59], fair learning [72], supervised learning [61, 49, 51, 74], adversarial deep learning [75], game theory [54, 58], robust optimization [4, 3], distributionally robust learning [46, 75, 24], meta-learning [70] and multi-agent reinforcement learning [15]. Many of these applications can be reformulated in

---

[*]Corresponding Author

38th Conference on Neural Information Processing Systems (NeurIPS 2024).

the following minimax form:

$$\min_{x\in\mathbb{R}^{d_1}} \max_{y\in\mathbb{R}^{d_2}} f(x,y), \tag{1}$$

where $f : \mathbb{R}^{d_1} \times \mathbb{R}^{d_2} \to \mathbb{R}$ is a smooth function, i.e., differentiable with a Lipschitz gradient; $f$ can possibly be nonconvex in $x$ and nonconcave in $y$. First-order primal-dual (FOPD) methods have been the leading computational approach for computing low-to-medium-accuracy stationary points for these problems because of their cheap iterations and mild dependence of their overall complexities on the problem dimension and data size [9, 8, 36]. In the context of FOPD methods, there are two key settings for (1):

(i) the *deterministic* setting, where the partial gradients $\nabla_x f$ and $\nabla_y f$ are exactly available,

(ii) the *stochastic* setting, where we have only access to (inexact) stochastic estimates of the partial gradients, in which case the problem in (1) is called a *stochastic minimax problem.*

It can be argued that the stochastic setting is more relevant to modern machine learning applications where gradients are typically estimated randomly from mini-batches of data, or sometimes intentionally perturbed with random noise to ensure data privacy [14, 34, 1].

**Convex and nonconvex minimax optimization.** In the convex case (when $f$ is convex[2] in $x$ and concave in $y$), several approaches have been considered including Variational Inequalities (VIs) and primal-dual algorithms, see. e.g. [29, 20, 5, 67, 12, 73, 60, 11] and the references therein. One disadvantage of using the VI approach for solving minimax problems (by identifying the *signed gradient map* $G(x,y) \triangleq [\nabla_x f(x,y)^\top, -\nabla_y f(x,y)^\top]^\top$ as the corresponding operator in the VI) is that one needs to set the primal and dual stepsize to be the same. This can be restrictive in applications where $f$ exhibits different smoothness properties in the primal ($x$) and dual ($y$) block coordinates –this is often the case in distributionally robust learning [73], adversarial learning [43] and in the Lagrangian reformulations of constrained optimization problems that involve many constraints [47]. The gap function $\mathcal{G}(x_k, y_k) \triangleq \sup_{x,y} f(x_k, y) - f(x, y_k)$, and the squared distance to the set of saddle points $\mathcal{D}(x_k, y_k) \triangleq \min\{\|x_k - x^\star\|^2 + \|y_k - y^\star\|^2 \mid (x^\star, y^\star) \text{ is a saddle point}\}$ are standard metrics for assessing the quality of the output $z_k = (x_k, y_k)$ generated by an FOPD algorithm after $k$ iterations among many others [35, 73, 20].

In the nonconvex setting, i.e., when $f$ is nonconvex in $x$, the aim is to compute a stationary point. Let $\mathcal{M}(x_k, y_k)$ denote a measure for the *stationarity* of iterates $(x_k, y_k)$; a common metric is the norm of the gradient, i.e., $\mathcal{M}(x_k, y_k) \triangleq \|\nabla f(x_k, y_k)\|$ and its variants such as $\|\nabla\Phi(x_k)\|$ when $f$ is strongly concave in $y$, where $\Phi(\cdot) = \max_y f(\cdot, y)$ denotes the primal function –for other metrics and relation between them, see [62]. There are several algorithms that admit (gradient) complexity guarantees for computing a stationary point of nonconvex minimax problems under various strong concavity, concavity or weak concavity-type assumptions in the $y$ variable –see the references in [62].

In this paper, we consider smooth nonconvex-PL (NCPL) problems where $f$ is a smooth function such that it is possibly nonconvex in $x$ and it satisfies the Polyak-Lojasiewicz (PL) condition in $y$. The PL condition is a weaker assumption (milder condition) than strong concavity in $y$ –in fact, PL condition in the dual does not even require quasi-concavity. NCPL problems constitute a rich class of problems arising in many ML applications including but not limited to fair classification [48], robust neural network training with dual regularization [48, eqn. (14)], overparametrized systems and neural networks [42], linear quadratic regulators [17], smoothed Lasso problems [23] subject to constraints, distributionally robust learning with $\ell_2$ regularization in the dual [72], deep AUC maximization [68] and covariance matrix learning with Wasserstein GANs [52]. For deterministic NCPL problems, the alternating gradient descent ascent (AGDA) method and its smoothed version (smoothed AGDA) have the complexity of $\mathcal{O}(\kappa^2/\epsilon^2)$ and $\mathcal{O}(\kappa/\epsilon^2)$, respectively, for finding a point $(\tilde{x}, \tilde{y})$ satisfying $\|\nabla f(\tilde{x}, \tilde{y})\| \leq \epsilon$ as shown in [70, 66]. Here $\kappa \triangleq \ell/\mu$ is the condition number, where $\ell$ is the Lipschitz constant of the gradient, and $\mu$ is the PL constant. For Catalyst-AGDA, [66] shows also the rate $\mathcal{O}(\kappa/\epsilon^2)$ for deterministic NCPL problems.

**In expectation and high-probability bounds.** Most of the existing guarantees in the literature for *stochastic* FOPD algorithms are provided in expectation, i.e., a bound on the number of iterations $k$ (or the stochastic gradient evaluations) is provided for $\mathbb{E}[\mathcal{G}(x_k, y_k)] \leq \varepsilon$ or $\mathbb{E}[\mathcal{D}(x_k, y_k)] \leq \varepsilon$ to hold (see, e.g., [73, 72, 29] and the references therein). Yet having such guarantees *on average* does

---

[2]$\hat{f} : \mathbb{R}^d \to \mathbb{R} \cup \{+\infty\}$ is called (merely) convex, if $\hat{f}(tx_1(1-t)x_2) \leq t\hat{f}(x_1) + (1-t)\hat{f}(x_2)$ for every $x_1, x_2 \in \mathbb{R}^d$ and $t \in [0,1]$ with the convention that $\alpha \leq +\infty$ for all $\alpha \in \mathbb{R}$.

not allow to control tail events, i.e., even if $\mathbb{E}[\mathcal{G}(x_k, y_k)]$ is small, $\mathcal{G}(x_k, y_k)$ can still be arbitrarily large with a non-zero probability. To this end, high-probability guarantees have been considered in the literature [35, 64, 20, 32, 22]. These results allow to control the risk associated with the worst-case tail events as they specify how many iterations would be sufficient to ensure $\mathcal{G}(x_k, y_k)$ is sufficiently small for any given failure probability $\bar{q} \in (0, 1)$. To derive high-probability bounds, one common approach involves running the algorithm in parallel multiple times and strategically selecting an optimal output to convert in-expectation bounds into high-probability guarantees [62, 37]. Alternatively, advanced concentration inequalities can be employed under light-tail assumptions to control noise accumulation across iterates without requiring multiple runs [53, 27]. For saddle point problems, we note that existing high-probability bounds mostly apply to the monotone VI setting, or to strongly convex/strongly concave (SCSC) minimax problems. To our knowledge, high-probability guarantees for nonconvex minimax problems are non-existent in the literature, even for nonconvex/strongly concave (NCSC) problems with the exception of trivial loose bounds one can obtain by a standard application of Markov's inequality (see Remark 13 for details). In particular, we note that the existing VI literature with high-probability bounds on non-monotone operators such as star-co-coercive operators [20, 55], do not apply to NCSC problems.[3]

**New high-probability bounds for NCPL optimization.** To address these shortcomings, we focus on developing high-probability guarantees for NCPL problems. Among the existing algorithms in the stochastic NCPL setting [64], stochastic gradient descent ascent (SGDA) methods and their variants are quite popular for ML applications, e.g., training GANs and adversarial learning, as SGDA is easy to implement due to its single-loop structure. Guarantees in expectation for stochastic NCSC problems are well supported by the literature - see [72, 63, 40, 6, 31, 30, 44, 66, 33, 39] and the references therein. To our knowledge, among single-loop methods for NCPL problems, the best guarantees in expectation are given by the *smoothed* alternating gradient descent ascent (`sm-AGDA`) method [66], which can compute an almost stationary point $(\tilde{x}, \tilde{y})$ satisfying $\mathbb{E}[\|\nabla f(\tilde{x}, \tilde{y})\|] \le \varepsilon$ in $\mathcal{O}(\ell \kappa^2 \delta^2 / \varepsilon^4 + \ell \kappa / \varepsilon^2)$ stochastic gradient calls, where $\delta^2$ is an upper bound on the variance of the stochastic gradients. In this work, we consider the `sm-AGDA` algorithm, and to our knowledge we provide the first-time high-probability bounds (using a single-loop method that does not resort to restarts and parallel runs) for the minimax problem (1) in the NCSC and NCPL settings. More precisely, we focus on a purely stochastic regime in which data streams over time which renders the use of mini-batch schemes or running the method in parallel impractical; therefore, approaches based on Markov's inequality [62] are no longer applicable (see also Remark 13).

**Contributions.** Our contributions are threefold:

- We present the first *high-probability* complexity result for the `sm-AGDA` algorithm in the NCPL setting by building upon a Lyapunov function first introduced in [70] for nonconvex-concave problems. Later, for the same Lyapunov function, state-of-the-art complexity bounds in *expectation* are provided for the NCPL setting in [66]. In this paper, we derive a novel descent property for this Lyapunov function in the almost sure sense (Theorem 7 and Corollary 8), allowing us to develop useful concentration arguments for it to derive high-probability bounds. Our Lyapunov analysis not only sheds light on the convergence properties of `sm-AGDA`, but also guides the parameter selection for `sm-AGDA`. Specifically, we show that `sm-AGDA` can compute an almost stationary point $(\tilde{x}, \tilde{y})$ satisfying $\|\nabla f(\tilde{x}, \tilde{y})\| \le \varepsilon$ with probability $1 - \bar{q} \in (0, 1)$ within $T_{\varepsilon, \bar{q}} = \mathcal{O}\left(\frac{\ell \kappa^2 \delta^2}{\varepsilon^4} + \frac{\kappa}{\varepsilon^2}\left(\ell + \delta^2 \log(1/\bar{q})\right)\right)$ stochastic gradient calls. The lower complexity bound of $\Omega(\frac{1}{\varepsilon^2} + \frac{1}{\varepsilon^4})$ for NCSC problems [38, 71] in expectation (see also [63, 72]) suggests that our high-probability bound for `sm-AGDA` is tight in terms of its dependence on $\varepsilon$. Furthermore, to our knowledge, these are the first high-probability guarantees for any algorithm in the NCPL setting.
- Under light-tail (sub-Gaussian) assumption on the gradient noise (Assumption 3), which is common in the literature [35, 32, 19], we develop a new concentration result (Theorem 9) that can be of independent interest. From this concentration inequality, we observe that the cost of strengthening the existing complexity result in expectation to a high-probability one is relatively low, i.e., in the final complexity, the probability parameter $\bar{q}$ *only* appears in an additive term that scales with $\varepsilon^{-2}$. Consequently, this represents a non-dominant overhead compared to the $\varepsilon^{-4}$ term already present in state-of-the-art expectation bounds [66].

---

[3]For example, when $\nabla f$ is smooth (Lipschitz and continuously differentiable) and the signed gradient map $G(x, y)$ is star-cocoercive with constant $\ell > 0$ around a stationary point $(x_*, y_*)$, then by [21, Lemma C.6], the operator $\text{Id} - \frac{2}{\ell} G(x, y)$ is non-expansive around $(x_*, y_*)$; thus, the Jacobian of $G$ at $(x_*, y_*)$ has non-negative eigenvalues implying $f$ is merely convex/merely concave around $(x_*, y_*)$, i.e., $f$ cannot be NCSC.

| Algorithm | Complexity | Problem | Metric | NC? |
|---|---|---|---|---|
| Epoch-GDA [64]$^\dagger$ | $\mathcal{O}\left(\frac{\delta^2}{\mu_s \varepsilon}\log(1/\bar{q})\right)$ | SCSC | $\mathcal{G}(\bar{z}_k)$ | ✗ |
| Clipped-SGDA [20]$^\ddagger$ | $\tilde{\mathcal{O}}\left(\max\left\{\frac{\ell}{\mu_s}, \frac{\delta^2}{\mu_s \varepsilon}\right\}\log\left(\frac{1}{\varepsilon}\right)\log\left(\kappa/\bar{q}\right)\right)$ | SCSC | $\mathcal{D}(z_k)$ | ✗ |
| Clipped-SEG [20]$^\flat$ | $\tilde{\mathcal{O}}\left(\max\left\{\frac{\ell}{\mu_s}, \frac{\delta^2}{\mu_s \varepsilon}\right\}\log\left(\frac{1}{\varepsilon}\right)\log\left(\kappa/\bar{q}\right)\right)$ | SCSC | $\mathcal{D}(z_k)$ | ✗ |
| Stochastic APD [35]$^{\triangleright *}$ | $\mathcal{O}\left(\frac{\ell\log\left(\frac{1}{\varepsilon}\right)}{\mu_s} + \frac{(1+\log(1/\bar{q}))\delta^2\log(1/\varepsilon)}{\mu_s\varepsilon}\right)$ | SCSC | $\mathcal{D}(z_k)$ | ✗ |
| Mirror-Prox [32]$^*$ | $\mathcal{O}\left(\max\left(\frac{\ell D^2}{\varepsilon^2}, \frac{\sigma^2 D^2}{\varepsilon^2}\right)\log\left(1/\bar{q}\right)\right)$ | MCMC | $\mathcal{G}(\bar{z}_k)$ | ✗ |
| Clipped-SGDA [20]$^\sharp$ | $\tilde{\mathcal{O}}\left(\max\{\frac{1}{\varepsilon}, \frac{\delta^2 R^2}{\varepsilon^2}\}\log\left(1/\bar{q}\right)\right)$ | MCMC | $\mathcal{G}_R(\bar{z}_k)$ | ✗ |
| Clipped-SEG [20] | $\tilde{\mathcal{O}}\left(\max\left(\frac{\ell R^2}{\varepsilon}, \frac{\sigma^2 R^2}{\varepsilon^2}\right)\log\left(1/\bar{q}\right)\right)$ | MCMC | $\mathcal{G}_R(\bar{z}_k)$ | ✗ |
| sm-AGDA [Our Paper, Coro. 14]$^*$ | $\mathcal{O}\left(\frac{\ell\kappa^2\delta^2}{\varepsilon^4} + \frac{\kappa}{\varepsilon^2}\left(\ell + \delta^2\log(1/\bar{q})\right)\right)$ | NCPL | $\frac{1}{k+1}\sum_{j=0}^{k}\|\nabla f(z_j)\|^2$ | ✓ |

Table 1: Summary of the high-probability bounds for minimax problem classes when the gradient of $f$ is Lipschitz (with parameter $\ell$) and stochastic gradient variance is bounded by $\delta^2$. The second column reports the complexity (number of calls to stochastic gradient oracle) required to achieve the (stationarity) metric reported in the fourth column to be at most $\varepsilon$ with probability $1 - \bar{q} \in (0,1)$; $\tilde{O}(\cdot)$ ignores some logarithmic terms. Here, $\mu_s$ is the strong convexity constant, $\mu$ is the PL constant, and $\kappa \triangleq \ell/\mu$. Let $G(z) \triangleq [\nabla_x f(z)^\top, -\nabla_y f(z)^\top]^\top$ with $z = (x,y)$ and $\bar{z}_k = \frac{1}{K+1}\sum_{j=0}^{K} z_j$, $\mathcal{G}(z) \triangleq \max_{\{\tilde{z} \in Z\}}\langle G(z), \tilde{z} - z_*\rangle$, where $Z$ is the domain of the problem with diameter $D \in (0, +\infty]$, and $\mathcal{G}_R(z) \triangleq \max_{\{\tilde{z} \in Z: \|z - z_*\| \leq R\}}\langle G(z), \tilde{z} - z_*\rangle$ where $z_* = (x_*^\top, y_*^\top)^\top$ is a stationary point. The third column reports the minimax problem class. The fifth column indicates whether the results supports nonconvexity, i.e., whether $f$ can be a smooth function nonconvex in $x$. $^\dagger$ [64] is a two-loop method. $^\ddagger$ Applicable to quasi-strongly monotone $G$ that is star-co-coercive around $z_*$ and supports heavy-tailed gradients. $^\flat$ Applicable to quasi-strongly monotone $G$ and supports heavy-tailed gradients. $^\triangleright$ Supports proximal steps to handle non-smooth convex penalty. $^\sharp$ Applies to monotone $G$ that is star-co-coercive around $z_*$. $^*$ Makes a light-tail assumption (Ass. 3).

- Third, we provide experiments that illustrate our theoretical results. We first provide an example of an NCPL-game with synthetic data and then focus on distributionally robust optimization problems with real data, illustrating the performance of the sm-AGDA in terms of high-probability guarantees.

## 2 Preliminaries and Technical Background

**Stationarity metric**. We consider the minimax problem in (1) for $f: \mathbb{R}^{d_1} \times \mathbb{R}^{d_2} \to \mathbb{R}$ such that $f$ is smooth (Assumption 1) and $f(x, \cdot)$ satisfies the PL property for all $x \in R^{d_1}$ (Assumption 2); moreover, we also assume that we only have access to unbiased stochastic estimates of $\nabla f$ such that the stochastic error $G(x, y, \xi) - \nabla f(x, y)$ has a light tail (Assumption 3) for any $(x, y)$, where $G(x, y, \xi)$ denote the stochastic estimate of $\nabla f(x, y)$ and $\xi$ denotes the randomness in the estimator.

Our aim is to compute a $(\varepsilon_x, \varepsilon_y)$-stationary point $(\tilde{x}, \tilde{y})$ for (1) such that $\|\nabla_x f(\tilde{x}, \tilde{y})\| \leq \varepsilon_x$ and $\|\nabla_y f(\tilde{x}, \tilde{y})\| \leq \varepsilon_y$. We also call $(\tilde{x}, \tilde{y})$ an $\varepsilon$-stationary point if $\|\nabla f(\tilde{x}, \tilde{y})\| \leq \varepsilon$. Clearly, whenever $(\tilde{x}, \tilde{y})$ is $(\varepsilon_x, \varepsilon_y)$-stationary, then it is also $\varepsilon$-stationary for $\varepsilon = (\varepsilon_x^2 + \varepsilon_y^2)^{1/2}$.

**Smoothed alternating gradient descent ascent** (sm-AGDA): The method can be considered as an *inexact* proximal point method and was introduced in [70]. More specifically, in each iteration of sm-AGDA, given a proximal center $z_t$ and the current iterate $(x_t, y_t)$, the method computes the next iterate $(x_{t+1}, y_{t+1})$ using a stochastic gradient descent ascent step on a regularized function $\hat{f}$:

$$\hat{f}(x, y; z_t) \triangleq f(x, y) + \frac{p}{2}\|x - z_t\|^2. \tag{2}$$

Following the stochastic alternating gradient descent ascent (stochastic AGDA) steps, the *proximal center* at iteration $t$, i.e., $z_t$, is updated as shown in Algorithm 1, where $G_x(x_t, y_t, \xi_{t+1}^x)$ and $G_y(x_{t+1}, y_t, \xi_{t+1}^y)$ denote conditionally unbiased stochastic estimators of the gradients $\nabla_x f(x_t, y_t)$ and $\nabla_y f(x_{t+1}, y_t)$. Throughout the analysis we assume that $\nabla f$ is Lipschitz, which is standard in the study of first-order optimization algorithms for smooth minimax problems; see, e.g., [72, 73, 75, 63].

**Assumption 1.** *(Lipschitz gradient) For all $(x_1, y_1), (x_2, y_2) \in \mathbb{R}^{d_1} \times \mathbb{R}^{d_2}$, there exists $\ell > 0$*

$$\|\nabla_x f(x_1, y_1) - \nabla_x f(x_2, y_2)\| \leq \ell(\|x_1 - x_2\| + \|y_1 - y_2\|) \tag{3}$$

$$\|\nabla_y f(x_1, y_1) - \nabla_y f(x_2, y_2)\| \leq \ell(\|x_1 - x_2\| + \|y_1 - y_2\|). \tag{4}$$

The following condition, known as Polyak-Łojaciewicz (PL) condition is weaker than assuming strong concavity in $y$, and does not even necessitate $f$ to be even quasi-concave in the $y$ variable. It holds in many ML applications including those in [48, 48, 42, 17, 23, 72, 68, 52, 66].

**Assumption 2.** *(PL condition in $y$) For every $x \in \mathbb{R}^{d_1}$, $\max_{y \in \mathbb{R}^{d_2}} f(x, y)$ has a non-empty solution set and a finite optimal value. Moreover, there exists $\mu > 0$ such that:*

$$\|\nabla_y f(x, y)\|^2 \geq 2\mu[\max_{y \in \mathbb{R}^{d_2}} f(x, y) - f(x, y)], \quad \forall x \in \mathbb{R}^{d_1}. \tag{5}$$

We assume that we have only access to stochastic estimates $G_x(x_t, y_t, \xi_{t+1}^x)$ and $G_y(x_{t+1}, y_t, \xi_{t+1}^y)$ of the partial gradients $\nabla_y f(x_k, y_k)$ and $\nabla_x f(x_{t+1}, y_t)$, where $\xi_{t+1}^x$ and $\xi_{t+1}^y$ are random variables defined on a probability space $(\Omega, \mathbb{P})$, i.e., the source of randomness in the gradient estimates. Note that sm-AGDA has Gauss-Seidel updates, i.e., the stochastic estimate of the partial gradient $G_y(x_{t+1}, y_t, \xi_{t+1}^y)$ is evaluated at the updated point $(x_{t+1}, y_t)$ instead of $(x_t, y_t)$. To capture

---

**Algorithm 1** sm-AGDA

**Input:** $(x_0, y_0, z_0)$, $\tau_1, \tau_2 > 0$, $\beta \in [0, 1]$, $p \geq 0$
**for** $t = 0, 1, 2, \ldots, T - 1$ **do**
$\quad x_{t+1} = x_t - \tau_1 [G_x(x_t, y_t, \xi_{t+1}^x) + p(x_t - z_t)]$
$\quad y_{t+1} = y_t + \tau_2 G_y(x_{t+1}, y_t, \xi_{t+1}^y)$
$\quad z_{t+1} = z_t + \beta(x_{t+1} - z_t)$
**end for**
**Output:** choose $(\tilde{x}, \tilde{y})$ uniformly from $\{(x_t, y_t)\}_{t=0}^{T-1}$

---

the sequential information flow, we next introduce the natural filtrations that represent all the information available before an update: Let $\xi_t^x$ and $\xi_t^y$ be revealed sequentially in the natural order of the sm-AGDA updates, i.e., $\xi_1^x \to \xi_1^y \to \xi_2^x \to \xi_2^y \to \xi_3^x \to \cdots$, and let $(\mathcal{F}_t^x)_{t \geq 1}$ and $(\mathcal{F}_t^y)_{t \geq 1}$ denote the associated filtration[4], i.e., let $\mathcal{F}_0^y \triangleq \{\emptyset, \Omega\}$, and

$$\mathcal{F}_{t+1}^x = \sigma(\mathcal{F}_t^y, \sigma(\xi_{t+1}^x)), \quad \mathcal{F}_{t+1}^y = \sigma(\mathcal{F}_{t+1}^x, \sigma(\xi_{t+1}^y)), \quad \forall\, t \geq 0. \tag{6}$$

Introducing multiple filtrations to represent the sequential information flow is common in the study of stochastic algorithms with Gauss-Seidel updates –see, e.g., papers on stochastic ADMM, and [7, 69]; and we follow the same approach. Consider the gradient noise (errors) at time $t \in \mathbb{N}$:

$$\Delta_t^x \triangleq G_x(x_t, y_t, \xi_{t+1}^x) - \nabla_x f(x_t, y_t), \quad \Delta_t^y \triangleq G_y(x_{t+1}, y_t, \xi_{t+1}^y) - \nabla_y f(x_{t+1}, y_t).$$

Finally, we also assume that the gradient noise is unbiased conditionally on the past information and that it admits a light (sub-Gaussian) tail.

**Assumption 3.** *(Light tail) For any $t \geq 0$, there exists scalars $\delta_x, \delta_y > 0$ such that*

$$\mathbb{E}\left[\Delta_t^x \mid \mathcal{F}_t^y\right] = 0, \quad \mathbb{P}\left[\|\Delta_t^x\| \geq s \mid \mathcal{F}_t^y\right] \leq 2e^{\frac{-s^2}{2\delta_y^2}}, \tag{7}$$

$$\mathbb{E}\left[\Delta_t^y \mid \mathcal{F}_{t+1}^x\right] = 0, \quad \mathbb{P}\left[\|\Delta_t^y\| \geq s \mid \mathcal{F}_{t+1}^x\right] \leq 2e^{\frac{-s^2}{2\delta_x^2}}. \tag{8}$$

For developing high-probability bounds in the learning context, it is common to assume that gradient estimates are sub-Gaussian [32, 35, 18]. While this assumption may not always hold (see e.g. [25, 56]), it often holds when gradients are estimated via mini-batching, as a consequence of the central limit theorem. It will also hold when the gradient noise is bounded. Additionally, adoption of differential privacy mechanisms within gradient-based schemes [14, 34, 1], to enhance data privacy, results frequently in sub-Gaussian gradient errors.

## 3 High-probability bounds for sm-AGDA

For analyzing sm-AGDA, similar to [66, 70], we consider the following Lyapunov function:

$$V_t \triangleq V(x_t, y_t; z_t) = \hat{f}(x_t, y_t; z_t) + 2P(z_t) - 2\Psi(y_t; z_t), \tag{9}$$

where $P(z)$ and $\Psi(\cdot; z)$ denote the saddle point value and the dual function value, respectively, of the auxiliary problem $\min_x \max_y \hat{f}(x, y; z)$ for any fixed $z$ and $\hat{f}$ defined in (2), i.e.,

$$\Psi(y; z) \triangleq \min_{x \in \mathbb{R}^{d_1}} \hat{f}(x, y; z) \quad \text{and} \quad P(z) \triangleq \min_{x \in \mathbb{R}^{d_1}} \max_{y \in \mathbb{R}^{d_2}} \hat{f}(x, y; z). \tag{10}$$

Next, we introduce a natural assumption, commonly made in the literature [66, 65]. Without this assumption, there are pathological cases where primal function $\Phi(x)$ may be unbounded leading to divergence of gradient-based methods; an example would be $f(x, y) = -x^2 - y^2$ in dimension one.

**Assumption 4.** *Consider the primal function $\Phi : \mathbb{R}^{d_1} \to \mathbb{R}$, i.e., $\Phi(x) = \max_{y \in \mathbb{R}^{d_2}} f(x, y)$. There exists $x^* \in \mathbb{R}^{d_1}$ such that $\Phi^* \triangleq \Phi(x^*) = \min_{x \in \mathbb{R}^{d_1}} \Phi(x)$.*

---

[4] Given a random variable $\xi$, $\sigma(\xi)$ denotes the $\sigma$-algebra generated by $\xi$; moreover, given two $\sigma$-algebras, $\Sigma_1$ and $\Sigma_2$, abusing the notation, $\sigma(\Sigma_1, \Sigma_2)$ denotes the $\sigma$-algebra generated by $\Sigma_1 \cup \Sigma_2$.

Under Assumption 4, it immediately follows that $V_t \geq \Phi^*$ for all $t \in \mathbb{N}$ –since $P(z) - \Psi(y, z) \geq 0$, $\hat{f}(x, y; z) - \Psi(y; z) \geq 0$ and $P(z) \geq \Phi^*$ for all $x, y, z$. We will next study the change $V_t - V_{t+1}$ in the Lyapunov function and show that an approximate descent property holds. First, we need two key lemmas that characterize the evolution of $\hat{f}(x_t, y_t; z_t)$ and $\Psi(y_t; z_t)$ over the iterations.

**Lemma 5.** *Suppose Assumptions 1, 2, 3 and 4 hold. Consider* `sm-AGDA` *given in Alg. 1 with $\tau_1 \in (0, \frac{1}{p+\ell}]$ and $\beta \in (0, 1]$. For any $t \in \mathbb{N}$, we have:*

$$
\begin{aligned}
\hat{f}(x_{t+1}, y_{t+1}; z_{t+1}) - \hat{f}(x_t, y_t; z_t) \leq & -\frac{\tau_1}{2} \|\nabla_x \hat{f}(x_t, y_t; z_t)\|^2 + \tau_2 \left(1 + \frac{\ell}{2}\tau_2\right) \|\nabla_y f(x_{t+1}, y_t)\|^2 \\
& + \tau_1((p+\ell)\tau_1 - 1)\langle \Delta_t^x, \nabla_x \hat{f}(x_t, y_t; z_t)\rangle + \frac{p+\ell}{2}\tau_1^2 \|\Delta_t^x\|^2 \\
& + \tau_2(1 + \ell\tau_2)\langle \nabla_y f(x_{t+1}, y_t), \Delta_t^y\rangle - \frac{p}{2\beta}\|z_t - z_{t+1}\|^2 + \frac{\ell\tau_2^2}{2}\|\Delta_t^y\|^2.
\end{aligned}
$$

*Proof.* The proof is provided in Appendix B.1. $\quad\square$

From Assumption 1, when $p > \ell$, the auxilliary function $\hat{f}(\cdot, y; z)$ is $(p - \ell)$-strongly convex for any fixed $y, z$; hence, there is a unique minimizer for every $y, z$ fixed, denoted by

$$
x^*(y, z) \triangleq \operatorname{argmin}_{x \in \mathbb{R}^{d_1}} \hat{f}(x, y; z), \tag{11}
$$

i.e., $\Psi(y, z) = \hat{f}(x^*(y, z), y; z)$. In the rest of the paper, we will take $p > \ell$ and exploit this property. The following lemma characterizes the change in the dual function $\Psi$.

**Lemma 6.** *Suppose Assumptions 1, 2, 3 and 4 hold. Consider the* `sm-AGDA` *iterate sequence $\{(x_t, y_t, z_t)\}_{t \in \mathbb{N}}$ for $p > \ell$. For any $t \in \mathbb{N}$, it holds that*

$$
\begin{aligned}
\Psi(y_{t+1}; z_{t+1}) - \Psi(y_t; z_t) \geq & \tau_2\langle \nabla_y f(x^*(y_t, z_t), y_t), \nabla_y f(x_{t+1}, y_t)\rangle + \tau_2\langle \nabla_y f(x^*(y_t, z_t), y_t), \Delta_t^y\rangle \\
& - \frac{L_\Psi}{2}\tau_2^2 \left(\|\nabla_y f(x_{t+1}, y_t)\|^2 + 2\langle \nabla_y f(x_{t+1}, y_t), \Delta_t^y\rangle + \|\Delta_t^y\|^2\right) \\
& + \frac{p}{2}\langle z_{t+1} - z_t, z_{t+1} + z_t - 2x^*(y_{t+1}, z_{t+1})\rangle,
\end{aligned}
$$

*where $L_\Psi \triangleq \ell\left(1 + \frac{p+\ell}{p-\ell}\right)$ and the map $x^*(\cdot, \cdot)$ is defined by (11).*

*Proof.* The proof is provided in Appendix B.2. $\quad\square$

The next result provides an approximate descent property on the Lyapunov function. Its proof builds on Lemmas 5 and 6 and a descent property on the function $P$ (given in Lemma 15 of the Appendix); and leverages smoothness properties of the functions $\hat{f}$ and $\Psi$ and the map $(y, z) \mapsto x^*(y, z)$ as well as the strong convexity of $\hat{f}$ with respect to $x$.

**Theorem 7.** *Suppose Assumptions 1, 2, 3 and 4 hold. Consider the* `sm-AGDA` *algorithm with parameters $p > \ell$, $\beta \in (0, 1]$, $\tau_1 \in (0, \frac{1}{p+\ell}]$ and $\tau_2 > 0$ chosen such that*

$$
c_0 \triangleq -\tau_2^2 \ell\nu + \tau_2\left(1 - \frac{\ell}{2}\tau_2 - L_\Psi\tau_2\right) \geq 0, \quad c_0' \triangleq \frac{p}{3\beta} - \left(\frac{2p^2}{p-\ell} + 48\beta\frac{p^3}{(p-\ell)^2}\right) \geq 0,
$$

*for some constant $\nu > 0$, where $L_\Psi = \ell\left(1 + \frac{p+\ell}{p-\ell}\right)$. Then,*

$$
\begin{aligned}
V_t - V_{t+1} \geq & c_1\|\nabla_x \hat{f}(x_t, y_t; z_t)\|^2 + c_2\|\nabla_y f(x^*(y_t, z_t), y_t)\|^2 + c_3\|x_t - z_t\|^2 \\
& + c_4\langle \nabla_x \hat{f}(x_t, y_t; z_t), \Delta_t^x\rangle + \langle c_5\nabla_y f(x_t, y_t) + c_6\nabla_y f(x^*(y_t, z_t), y_t), \Delta_t^y\rangle \quad (12) \\
& + c_7\|\Delta_t^x\|^2 + c_8\|\Delta_t^y\|^2,
\end{aligned}
$$

*for some constants $\{c_i\}_{i=1}^8 \subset \mathbb{R}$ that are explicitly given in Appendix C, which may depend on $\nu$, as well as the problem and* `sm-AGDA` *parameters that can be chosen such that $c_1, c_2, c_3 > 0$.*

*Proof.* The proof is given in Appendix C. $\quad\square$

With some specific choice of parameters in `sm-AGDA`, we can obtain simplifications to the coefficients $\{c_i\}_{i=1}^8$ from Theorem 7 (explicitly given in Appendix C). As such, this yields the following corollary.

**Corollary 8.** *Under the premise of Theorem 7, let $p = 2\ell$, $\tau_1 \in (0, \frac{1}{3\ell}]$, $\tau_2 = \frac{\tau_1}{48}$, $\beta = \alpha\mu\tau_2$ for $\alpha \in (0, \frac{1}{406}]$. Then, $\frac{\tilde{A}_{t+1}-\tilde{A}_t}{\tau_1} \leq -\tilde{B}_t + \tilde{C}_{t+1} + \tilde{D}_{t+1}$ for all $t \in \mathbb{N}$, where $\nu = \frac{12}{\tau_1\ell}$ and*
$$\tilde{A}_t \triangleq \tau_1 V_t, \quad \tilde{B}_t \triangleq \frac{\tau_1}{5}\|\nabla_x\hat{f}(x_t, y_t; z_t)\|^2 + \frac{\tau_2}{8}\|\nabla_y f(x^*(y_t, z_t), y_t)\|^2 + \frac{\beta p}{8}\|x_t - z_t\|^2,$$
$$\tilde{C}_{t+1} \triangleq \Big[\Big(192\beta p\Big(\frac{p+\ell}{p-\ell}\Big)^2\ell^2\tau_2^2 + \frac{4\ell}{\nu} + 4c_0\ell^2 + 2c_0'\beta^2\Big)\tau_1^2 + \Big((p+\ell)\tau_1 - 1\Big)\tau_1\Big]\langle\nabla_x\hat{f}(x_t, y_t; z_t), \Delta_t^x\rangle$$
$$+ \tau_2\langle(1 + \ell\tau_2 + 2L_\Psi\tau_2)\nabla_y f(x_t, y_t) - 2\nabla_y f(x^*(y_t, z_t), y_t), \Delta_t^y\rangle,$$
$$\tilde{D}_{t+1} \triangleq 2\ell\tau_1^2\|\Delta_t^x\|^2 + 8\ell\tau_2^2\|\Delta_t^y\|^2.$$

*Proof.* The proof is given in Appendix D. $\qquad\square$

Next, we provide a concentration inequality which will be key to obtain our high-probability bounds.

**Theorem 9.** *Let $\{\mathcal{F}_t\}_{t\in\mathbb{N}}$ be a filtration on $(\Omega, \mathcal{F}, \mathbb{P})$. Let $A_t, B_t, C_t, D_t$ be four stochastic processes adapted to the filtration such that there exist $\sigma_C, \sigma_D > 0$ and $\tau_1 > 0$ such that for all $t \in \mathbb{N}$: (i) $B_t \geq 0$, (ii) $\mathbb{E}[e^{\lambda C_{t+1}} \mid \mathcal{F}_t] \leq e^{\lambda^2\sigma_C^2 B_t}$ for all $\lambda > 0$, (iii) $\mathbb{E}[e^{\lambda D_{t+1}} \mid \mathcal{F}_t] \leq e^{\lambda\sigma_D^2}$ for all $\lambda \in [0, \frac{1}{\sigma_D^2}]$ and (iv) $\frac{A_{t+1}-A_t}{\tau_1} \leq -B_t + C_{t+1} + D_{t+1}$. Then, for any $\bar{q} \in (0, 1]$, we have*
$$\mathbb{P}\left(\frac{\tau_1}{2}\sum_{t=0}^{T-1} B_t \leq (A_0 - A_T) + \tau_1\sigma_D^2 T + 2\tau_1\max\{2\sigma_C^2, \sigma_D^2\}\log\Big(\frac{1}{\bar{q}}\Big)\right) \geq 1 - \bar{q}.$$

*Proof.* The proof is provided in Appendix E. $\qquad\square$

**Remark 10.** *While the above concentration inequality seems tailored to the analysis of sm-AGDA, it can also aid in deriving high probability bounds for many other first-order methods for nonconvex minimax problems that outputs a randomized iterate; indeed, the majority of existing Lyapunov arguments in the nonconvex setting are built upon constructing telescoping sums in line with Theorem 9, e.g., stochastic alternating GDA [66] for NCPL minimax problems and optimistic GDA [45] for strongly convex-strongly concave problems.*

We next present our main result which provides a high-probability bound on the `sm-AGDA` iterates. The main idea of the proof is to apply Theorem 9 to the processes introduced in Corollary 8.

**Theorem 11.** *In the premise of Corollary 8, `sm-AGDA` iterates $(x_t, y_t)$ for $\tau_1 \leq \frac{1}{3\ell}$ satisfy*
$$\mathbb{P}\left(\frac{1}{T}\sum_{t=0}^{T-1}\big[\|\nabla_x f(x_t, y_t)\|^2 + \kappa\|\nabla_y f(x_t, y_t)\|^2\big] \leq \mathcal{Q}_{\bar{q}, T},\right) \geq 1 - \bar{q}, \quad \forall\, T \in \mathbb{N}, \quad \forall\, \bar{q} \in (0, 1],$$
*for some $\mathcal{Q}_{\bar{q}, T} = \mathcal{O}\Big(\frac{\kappa(\Delta_0 + b_0)}{\tau_1 T} + \kappa(\delta_x^2 + \delta_y^2)\Big(\tau_1\ell + \frac{1}{T}\log\big(\frac{1}{\bar{q}}\big)\Big)\Big)$ explicitly stated in Appendix F, where $\Delta_0 \triangleq \Phi(z_0) - \Phi^*$, $b_0 \triangleq 2\sup_{x,y}\{\hat{f}(x_0, y; z_0) - \hat{f}(x, y_0; z_0)\}$.*

*Proof Sketch.* Let the stochastic processes $A_t, B_t, C_t, D_t$ in Theorem 9 be chosen as $A_t = \tilde{A}_t$, $B_t = \tilde{B}_t$, $C_t = \tilde{C}_t$, $D_t = \tilde{D}_t$ where $\tilde{A}_t, \tilde{B}_t, \tilde{C}_t, \tilde{D}_t$ are defined in Corollary 8 and $\tau_1 > 0$ be the primal stepsize in `sm-AGDA`; according to Corollary 8, we have $\frac{A_{t+1}-A_t}{\tau_1} \leq -B_t + C_{t+1} + D_{t+1}$ for $t \in \mathbb{N}$. Since $\Delta_t^x$ and $\Delta_t^y$ admit sub-Gaussian tails, it can be shown that the conditions of Theorem 9 are satisfied for some appropriate constants $\sigma_C^2$ and $\sigma_D^2$. Therefore, Theorem 9 implies a tail bound on $\sum_{t=0}^{T-1}\tilde{B}_t$. Using the relation between $f$ and $\hat{f}$, one can also show that $\|\nabla_x f(x_t, y_t)\|^2 + \kappa\|\nabla_y f(x_t, y_t)\|^2 = \mathcal{O}(\tilde{B}_t)$, for all $t \in \mathbb{N}$. This last inequality allows to translate the tail bound for $\sum_{t=0}^{T-1}\tilde{B}_t$ to a tail bound for $\sum_{t=0}^{T-1}\|\nabla_x f(x_t, y_t)\|^2 + \kappa\|\nabla_y f(x_t, y_t)\|^2$. The details of the proof is provided in Appendix F of the supplementary material. $\qquad\square$

**Remark 12.** *Suppose `sm-AGDA`, given in Alg. 1, is run for $T$ iterations, and it outputs a randomly selected iterate $(x_U, y_U)$, where the random iteration index $U$ is chosen uniformly at random from the set $\{0, 1, \ldots, T-1\}$, i.e., $\mathbb{P}(U = t) = 1/T$ for $t = 0, 1, \ldots, T-1$. Theorem 11 implies that*
$$\mathbb{P}\left(\|\nabla_x f(x_U, y_U)\|^2 + \kappa\|\nabla_y f(x_U, y_U)\|^2 \leq \mathcal{Q}_{\bar{q}, T}\right) \geq 1 - \bar{q}.$$

*Furthermore, in comparison with existing complexity bounds in expectation for `sm-AGDA` [66], our quantile bound requires only an overhead of order $\mathcal{O}(\varepsilon^{-2}\log(1/\bar{q}))$. Unless $\bar{q}$ is very small, this is typically negligible in comparison to the $\mathcal{O}(\varepsilon^{-4})$ already present in rates in expectation.*

**Remark 13.** *In contrast to high-probability bounds derived from standard Markov-type arguments, our approach achieves significantly better scaling with respect to both $\bar{q}$ and $\epsilon$. Specifically, consider an oracle that can generate a sample $(\hat{x}, \hat{y})$ with $\mathbb{E}[\|\nabla f(\hat{x}, \hat{y})\|] \leq \epsilon$ after $\mathcal{G}(\epsilon)$ iterations/stochastic samples. In particular, [66] shows that one can take $\mathcal{G}(\epsilon) = \mathcal{O}\left(\frac{\ell \kappa^2 \delta^2}{\epsilon^4} + \frac{\kappa \ell}{\epsilon^2}\right)$ for the* `sm-AGDA` *algorithm assuming the variance of the stochastic gradient is bounded by $\delta^2$. A naive high-probability bound could be constructed by ensuring $\mathbb{E}[\|\nabla f(\hat{x}, \hat{y})\|] \leq \bar{q} \cdot \epsilon$ and applying Markov's inequality to yield an $\epsilon$-stationary point with probability at least $1 - \bar{q}$. However, this approach results in a complexity bound of $\mathcal{G}(\bar{q}\epsilon) = O\left(\frac{\ell \kappa^2 \delta^2}{\bar{q}^4 \epsilon^4} + \frac{\kappa \ell}{\bar{q}^2 \epsilon^2}\right)$, leading to a significantly worse dependence on $\bar{q}$ than ours. Alternatively, following the rationale in [19, 62], to generate a high-probability bound, one can run the* `sm-AGDA` *algorithm $m = \Omega(\log(1/\bar{q}))$ times in parallel; where in each run we generate an $\varepsilon/2$-solution and among the solutions, we select the one with the smallest estimated gradient norm. This would require $m\mathcal{G}(\epsilon) = \mathcal{O}\left(\log(1/\bar{q})\frac{\ell \kappa^2 \delta^2}{\epsilon^4} + \log(1/\bar{q})\frac{\kappa \ell}{\epsilon^2}\right)$ iterations/stochastic samples. In this approach, the logarithmic term $\log\left(\frac{1}{\bar{q}}\right)$ multiplies the high-order $\mathcal{O}(\frac{1}{\varepsilon^4})$ term, whereas in our approach it only affects the second-order $\mathcal{O}(\frac{1}{\varepsilon^2})$ term. Therefore, our results scale better with respect to $\bar{q}$ and $\varepsilon$. In addition, such a (multiple) parallel run approach, is often impractical in streaming/online settings, where data arrives sequentially, and real-time processing is essential.*

**Corollary 14.** *Under the premise of Theorem 11, consider running the* `sm-AGDA` *method for some fixed number of iterations $T \in \mathbb{N}$ with parameters chosen as $\tau_1 = \min\left(\frac{1}{3\ell}, \frac{48\sqrt{\Delta_0 + b_0}}{\sqrt{T\ell\delta^2}}\right)$ and $\tau_2 = \tau_1/48$ where $\delta^2 \triangleq \delta_x^2 + \delta_y^2$. Then, for any $\bar{q} \in (0, 1)$,* `sm-AGDA` *can compute an $(\varepsilon, \varepsilon/\sqrt{\kappa})$ stationarity point with probability at least $1 - \bar{q}$ when the number of iterations $T$ is fixed to*
$$T_{\varepsilon, \bar{q}} = \mathcal{O}\left(\frac{(\Delta_0 + b_0)\ell\kappa}{\varepsilon^2} + \frac{\delta^2 \log\left(\frac{1}{\bar{q}}\right)\kappa}{\varepsilon^2} + \frac{\delta^2(\Delta_0 + b_0)\ell\kappa^2}{\varepsilon^4}\right)$$ *which requires $T_{\varepsilon, \bar{q}}$ stochastic gradient calls.*

*Proof.* This is a direct consequence of Theorem 11, a proof is provided in Appendix G. $\qquad\square$

## 4 Numerical Illustrations

In this section, we illustrate the performance of `sm-AGDA`. We consider an NCPL problem with synthetic data, as well as a nonconvex DRO problem using real datasets. For synthetic experiments, we used an ASUS Laptop model Q540VJ with 13th Generation Intel Core i9-13900H using 16GB RAM and 1TB SSD hard drive. For the DRO experiments, we used a high-performance computing cluster with automatic GPU selection (NVIDIA RTX 3050, RTX 3090, A100, or Tesla P100) based on GPU availability, ensuring optimal use of computational resources.

**Synthetic experiments on an NCPL game.** We consider the following NCPL problem:

$$\min_{x \in \mathbb{R}^{d_1}} \max_{y \in \mathbb{R}^{d_2}} m_1\left[\|x\|^2 + \sin\left(3\sqrt{\|x\|^2 + 1}\right)\right] + x^\top K y - m_2\left[\|y\|^2 + 3\sin^2(\|y\|)\right], \qquad (13)$$

which can be interpreted as a game between two players [48, 35] where $m_1, m_2 > 0$ are constants and the symmetric matrix $K$ is set randomly, similar to the standard bilinear game setting considered in [35]. More specifically, we set $K = 10\tilde{K}/\|\tilde{K}\|$, $\tilde{K} = (M + M^\top)/2$ where $M$ is a $d \times d$ matrix with entries being i.i.d centered Gaussian having variance $\sigma^2$. This problem is nonconvex in $x$ (without satisfying the PL condition in $x$). Though the exact gradient is known, we consider a stochastic gradient oracle, which returns *noisy* gradients similar to the setting of [35, 11, 16, 2], i.e., for each iteration $t \in \{0, ..., T-1\}$, $G_x(x_t, y_t; \xi_{t+1}^x) = \nabla_x f(x_t, y_t) + \xi_{t+1}^x$ and $G_y(x_{t+1}, y_t, \xi_{t+1}^y) = \nabla_y f(x_{t+1}, y_t) + \xi_{t+1}^y$, with $(\xi_{t+1}^x)_{t \geq 0} \overset{iid}{\sim} \mathcal{N}(\mathbf{0}, \delta^2 I_{d_1})$ and $(\xi_{t+1}^y)_{t \geq 0} \overset{iid}{\sim} \mathcal{N}(\mathbf{0}, \delta^2 I_{d_2})$ where $I_d$ is the $d \times d$ identity matrix and $\delta^2$ is some constant variance. This setting satisfies all our assumptions, and our high-probability results (Theorem 11 and Coro. 14) are applicable. In this experiment, we fix $d_1 = d_2 = 30$, and $m_1 = m_2 = \delta^2 = \sigma^2 = 1$. The solution to this problem is $(x^*, y^*) = (\mathbf{0}, \mathbf{0})$.

*Experimental results.* The parameters of the problem are explicitly available as $\mu = 2m_2$, and $\ell = \max\{12m_1, 8m_2, \|K\|\}$. To illustrate Theorem 11, we set $\beta = \frac{\tau_2\mu}{1600}, \tau_2 = \frac{\tau_1}{48}, p = 2\ell$ and we considered two cases: $\tau_1 = \frac{1}{3\ell}$ (long step) and $\tau_1 = \frac{1}{12\ell}$ (short step) to explore the behavior of `sm-AGDA` for different stepsizes. We generated $N = 25$ sample paths for $T = 10,000$ iterations, and on the left panel of Fig. 1, for each iteration $t$, we report the average of $\mathcal{M}_\kappa(t) \triangleq \|\nabla_x f(x_t, y_t)\|^2 + \kappa\|\nabla_y f(x_t, y_t)\|^2$ over $N = 25$ realizations corresponding to different sample paths, and the shaded region depicts the range statistic, i.e., for every fixed iteration $t$, we shade the

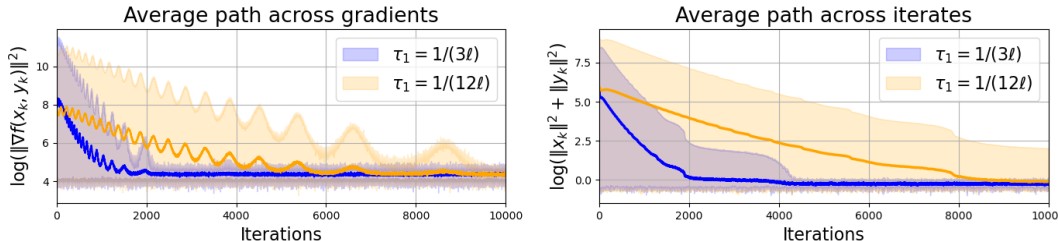

Figure 1: NCPL game with $\tau_1 = \frac{1}{3\ell}$ (long step) and $\tau_1 = \frac{1}{12\ell}$ (short step). **(Left)** Average of $\mathcal{M}_k(t)$ over 25 sample paths vs. iterations $t$. **(Right)** Average of $\mathcal{I}(t)$ over 25 sample paths vs. iterations $t$. In both plots, shaded regions depict the range statistic over 25 sample paths.

vertical line between the maximum and minimum values of $\mathcal{M}_\kappa(t)$ for the 25 paths. On the right panel of Fig. 1, we also report the squared distance, $\mathcal{I}(t) \triangleq \|x_t - x^*\|^2 + \|y_t - y^*\|^2$, in a similar manner. We observe that the range statistic for $\mathcal{M}_\kappa(t)$ diminishes to a value inversely proportional to $T$ as $t \to T$; this is inline with our theoretical results in Theorem 11. The existence of sinusoidal terms in the minimax objective is a source for oscillatory behavior in these figures. As $t \to T$, both $\mathcal{M}_\kappa(t)$ and $\mathcal{I}(t)$ exhibit oscillations, of which amplitude are smaller for the small step size compared to the large step-size –at the expense of a slower convergence; the iterate paths are not as oscillatory.

For $T = 10,000$ and $\tau_1 = \frac{1}{3\ell}$ fixed, we next consider the path averages $X_T \triangleq \frac{1}{T} \sum_{t=0}^{T-1} \mathcal{M}_\kappa(t)$. Indeed, based on 1000 sample paths, each for $T = 10,000$ iterations, we compare the empirical quantiles of $X_T$ with the theoretical upper bound $\mathcal{Q}_{q,T}$ on its quantiles (implied by Theorem 11). Figure 2 shows the cumulative distribution function (CDF) of the empirical distribution alongside the theoretical explicit upper bound $\mathcal{Q}_{q,T}$ for the stationarity measure $\|\nabla_x f(x_U, y_U) + \kappa \nabla_y f(x_U, y_U)\|$ with $U$

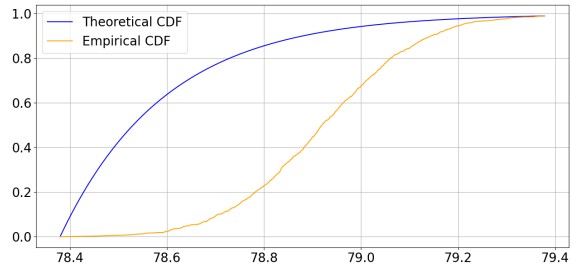

Figure 2: Comparison of our theoretical upper bound in Theorem 11 and the empirical stationarity measure of sm-AGDA for Problem (13). Results are reported as cumulative distribution functions.

uniform over $\{1, \ldots, T\}$. The details of the empirical quantile estimation are provided Appendix I due to space considerations. We observe that the empirical CDF has a sigmoid-like shape, while the theoretical quantiles that lie above display a concave form. This difference may arise because the theoretical quantile bounds $\mathcal{Q}_{q,T}$ are designed to capture the worst-case behavior across the class of NCPL problems, whereas this specific NCPL example may not represent the worst-case scenario.

**Distributionally Robust nonconvex Logistic Regression.** We consider the DRO problem

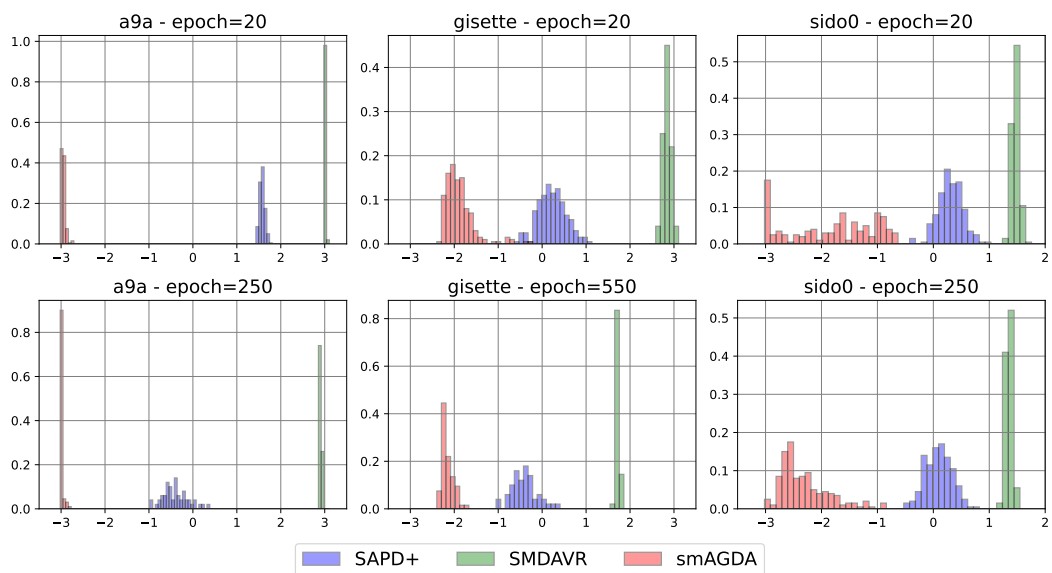

Figure 3: Histograms for the stationarity measure $\log_{10} \|\nabla f(x_t, y_t)\|^2$ for sm-AGDA and baseline algorithms (SAPD+, SMDA, and SMDAVR) on a9a, gisette, and sido0 datasets. Each algorithm is run 200 times. First row reports histograms after 20 epochs, and second row reports histograms at the end of training.

$$\min_{x \in \mathbb{R}^{d_1}} \max_{y \in Y} \Big( \frac{1}{d_2} \sum_{j=1}^{d_2} y_j \ell_j(x; a_j, b_j) + r(x) - g(y) \Big), \tag{14}$$

where $\ell_j(x; a_j, b_j) = \log\big(1 + \exp\big(-b_j \mathbf{a}_j^\top \mathbf{x}\big)\big)$ denotes the logistic loss tied to an input-output pair $(a_j, b_j) \in \mathbb{R}^{d_1} \times \{-1, 1\}$, and $r(x) = \lambda_1 \sum_{i=1}^{d_1} \frac{\omega x_i^2}{1 + \omega x_i^2}$ a primal regularization for the learning model $x \in \mathbb{R}^{d_1}$. We allow the distribution $y \in Y \triangleq \{y \in \mathbb{R}^{d_2} : y \geq 0, \mathbf{1}^\top y = 1\}$ to deviate from the uniform distribution $u \triangleq \frac{1}{d_2} \mathbf{1}$ where $\mathbf{1}$ denotes the vector of ones, and we penalize the distance between $y$ and $u$ through the regularization map $g : y \mapsto \frac{\lambda_2 d_2}{2} \|y - u\|^2$. We set the regularization parameters as $\omega = 10$, $\lambda_1 = 10e^{-4}$, and $\lambda_2 = 1$. Since $r$ is nonconvex with a Lipschitz gradient and $g$ is strongly convex, this is an NCSC problem.

*Datasets, Algorithms and Hyperparameters.* We consider three standard datasets for this problem, which are summarized as follows: The `sido0` dataset [50] has $d_1 = 4932$ and $d_2 = 12678$. The `gisette` dataset [26] has $d_1 = 5000$ and $d_2 = 6000$. Finally, the `a9a` dataset [13] has $d_1 = 123$ and $d_2 = 32561$. We compare the performances of `sm-AGDA` against two other baselines that achieve state-of-the-art performance in expectation for these datasets [72]. Specifically, we evaluate `SAPD+`, which is a two-loop method where the subproblems are solved by the `SAPD` algorithm [73], and `SMDAVR`, a variance reduced extension of `SMDA` algorithm [31]. Since (14) is constrained, we augment `sm-AGDA` with a projection step in the update of the y variable onto the $d_2$-dimensional simplex and adopt the analogous stationarity metric $\|\nabla_x f(x_t, y_t)\|^2 + \|P_Y \nabla_y f(x_t, y_t)\|^2$ for constrained problems where $P_Y$ is a projection to the dual domain $Y$. For all datasets, the primal stepsize $\tau_1$ of `sm-AGDA` is tuned via a grid-search over $\{10^{-k}, 1 \leq k \leq 4\}$. The dual stepsize $\tau_2$ is set as $\tau_2 = \frac{\tau_1}{48}$. Similarly, $\beta$ is estimated through a grid-search over $\{10^{-k}, 3 \leq k \leq 5\}$. The parameter $p$ is also tuned similarly on a grid, our code is provided as a supplementary document for the details. For other methods, our hyperparameters are tuned in accordance with [72].

*Experimental results.* In Figure 2, we plot histograms of our stationarity metric, across 200 runs in a logarithmic scale. We report the stationarity measure both in early phase of the training (i.e. $t = 20$ epochs), and in later phases (i.e. $t = 550$ epochs for `gisette` and $t = 250$ epochs for `a9a` and `sido0`). Our theoretical results are presented for unconstrained problems in the dual, therefore they are not directly applicable to the DRO problem where the dual domain is constrained. That being said, we observe that they are still predictive of performance in the DRO setting. More specifically, Figure 2 is supportive of our high-probability complexity bounds for `sm-AGDA`, in the sense that the distribution of the stationarity metric for `sm-AGDA` tends to concentrate. Notably, it outperforms the concentration behaviour of the other baselines. Furthermore, we observe that histograms for all baselines hardly evolve after 20 epochs. This is consistent with previous experiments carried on these datasets [72] where performance was measured in terms of the decay of the average loss and its standard deviation. As such, we conclude that `sm-AGDA` performs better both in the early phase and the later stage. In our experience, we observed `sm-AGDA` could accomodate larger stepsizes compared to the other algorithms, which may have contributed to its good performance.

## 5 Conclusion

Existing high-probability bounds only apply to convex/concave minimax problems or non-monotone variational inequality problems under restrictive assumptions to our knowledge. We close this gap by providing the first high-probability complexity guarantees for nonconvex/PL minimax problems satisfying the PL-condition in the dual variable for the `sm-AGDA` method. We also provide numerical results for an NCPL example and for nonconvex distributionally robust logistic regression.

## Acknowledgements

Yassine Laguel, Yasa Syed and Mert Gürbüzbalaban's research are supported in part by the grants Office of Naval Research Award Numbers N00014-21-1-2244 and N00014-24-1-2628, National Science Foundation (NSF) CCF-1814888 and NSF DMS-2053485. Necdet Serhat Aybat's work was supported in part by the Office of Naval Research Awards N00014-21-1-2271 and N00014-24-1-2666.

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

# High-probability complexity guarantees
# for nonconvex minimax problems
## APPENDIX

The organization of the Appendix is as follows:

- In Section A, we summarize the key notations that are used throughout the main paper and the Appendix.
- In Section B, we provide the proofs of Lemmas from Section 3 that were used for deriving the approximate Lyapunov descent property.
- In Section C, we present the proof of Theorem 7 which provides an approximate descent property in terms of the Lyapunov function for the sm-AGDA iterates.
- In Section D, we provide the proof of Corollary 8, which studies particular choice of parameters within Theorem 7.
- In Section E we present a proof of Theorem 9 which provides a concentration inequality.
- In Section F, we provide the proof of Theorem 11 which yields high-probability results for the sm-AGDA algorithm.
- In Section G, we provide a proof of Corollary 14 that provides a high-probability (iteration) complexity bound for the sm-AGDA iterates.
- In Section H, we present supplementary lemmas that were used to derive the results.
- In Section I, we provide further details about numerical experiments.

## A   Notation

The key notations that will be used throughout the Appendix is as follows:

- $\hat{f}(x, y; z) = f(x, y) + \frac{p}{2}\|x - z\|^2$ denotes the auxiliary problem.
- $\Psi(y; z) = \min_x \hat{f}(x, y; z)$ is the dual function of the auxiliary problem.
- $\Phi(x; z) = \max_y \hat{f}(x, y; z)$ is the primal function of the auxiliary problem.
- $P(z) = \min_x \max_y \hat{f}(x, y; z)$ is the optimal value of the primal problem $\min_x \Phi(x; z)$
- $x^*(y, z) = \operatorname{argmin}_x \hat{f}(x, y; z)$ for given $y, z$ in the auxiliary function.
- $x^*(z) = \operatorname{argmin}_x \Phi(x; z)$ is the unique optimal solution to the auxiliary primal problem.
- $Y^*(z) = \operatorname{argmax}_y \Psi(y; z)$ is the set of optimal solutions to the auxiliary dual problem.
- $y^+(z) = y + \tau_2 \nabla_y f(x^*(y, z), y)$ denotes an update in $y$ in the direction of the gradient of the dual function, i.e., along the direction $\nabla \Psi(y; z) = \nabla_y f(x^*(y, z), y)$.
- $\hat{G}_x(x, y, \xi; z) \triangleq G_x(x, y, \xi) + p(x - z)$
- $\Delta_t^x = G_x(x_t, y_t, \xi_{t+1}^x) - \nabla_x f(x_t, y_t)$ and $\Delta_t^y = G_y(x_{t+1}, y_t, \xi_{t+1}^y) - \nabla_y f(x_{t+1}, y_t)$ denote the gradient error, i.e., the difference between the stochastic estimates of the partial gradients and the exact partial gradients.

## B   Proofs of Lemmas from Section 3

### B.1   Proof of Lemma 5

**Lemma 5.** *Suppose Assumptions 1, 2 and 3 hold. Consider sm-AGDA, stated in Algorithm 1, with $\tau_1 \in (0, \frac{1}{p+\ell}]$ and $\beta \in (0, 1]$. For any $t \in \mathbb{N}$, we have:*

$$\hat{f}(x_{t+1}, y_{t+1}; z_{t+1}) - \hat{f}(x_t, y_t; z_t)$$

$$\leq -\frac{\tau_1}{2}\|\nabla_x \hat{f}(x_t, y_t; z_t)\|^2 + \tau_2\left(1 + \frac{\ell}{2}\tau_2\right)\|\nabla_y f(x_{t+1}, y_t)\|^2 - \frac{p}{2\beta}\|z_t - z_{t+1}\|^2$$
$$+ \tau_1((p+\ell)\tau_1 - 1)\langle \Delta_t^x, \nabla_x \hat{f}(x_t, y_t; z_t)\rangle + \tau_2(1 + \ell\tau_2)\langle \Delta_t^y, \nabla_y f(x_{t+1}, y_t)\rangle$$
$$+ \frac{p+\ell}{2}\tau_1^2\|\Delta_t^x\|^2 + \frac{\ell\tau_2^2}{2}\|\Delta_t^y\|^2.$$

*Proof.* Since, $\hat{f}(\cdot, y; z)$ is $(p+\ell)$-smooth, we have

$$\hat{f}(x_{t+1}, y_t; z_t) - \hat{f}(x_t, y_t; z_t)$$
$$\leq \langle x_{t+1} - x_t, \nabla_x \hat{f}(x_t, y_t; z_t)\rangle + \frac{p+\ell}{2}\|x_{t+1} - x_t\|^2$$
$$= -\tau_1 \langle \hat{G}_x(x_t, y_t; z_t), \nabla_x \hat{f}(x_t, y_t; z_t)\rangle + \frac{p+\ell}{2}\tau_1^2\|\hat{G}_x(x_t, y_t; z_t)\|^2.$$
$$= (\frac{p+\ell}{2}\tau_1^2 - \tau_1)\|\nabla_x \hat{f}(x_t, y_t; z_t)\|^2 - (\tau_1 + (p+\ell)\tau_1^2)\langle \Delta_t^x, \nabla_x \hat{f}(x_t, y_t; z_t)\rangle$$
$$+ \frac{p+\ell}{2}\|\Delta_t^x\|^2,$$

where last equality follows from $\hat{G}_x(x_t, y_t; z_t) = \nabla_x \hat{f}(x_t, y_t; z_t) + \Delta_t^x$. Hence, for $\tau_1 \leq \frac{1}{p+\ell}$, we have

$$\hat{f}(x_{t+1}, y_t; z_t) - \hat{f}(x_t, y_t; z_t)$$
$$\leq -\frac{\tau_1}{2}\|\nabla_x \hat{f}(x_t, y_t; z_t)\|^2 + \tau_1((p+\ell)\tau_1 - 1)\langle \Delta_t^x, \nabla_x \hat{f}(x_t, y_t; z_t)\rangle + \frac{p+\ell}{2}\tau_1^2\|\Delta_t^x\|^2. \tag{15}$$

Similarly, we observe that for all, $\nabla_y \hat{f}(x, y; z) = \nabla_y f(x, y)$, for all $x, y, z$, which together with the smoothness of $f(x, \cdot)$ implies

$$\hat{f}(x_{t+1}, y_{t+1}; z_t) - \hat{f}(x_{t+1}, y_t; z_t)$$
$$\leq \langle \nabla_y \hat{f}(x_{t+1}, y_t; z_t), \ y_{t+1} - y_t\rangle + \frac{\ell}{2}\|y_{t+1} - y_t\|^2$$
$$= \tau_2 \langle \nabla_y f(x_{t+1}, y_t), G_y(x_{t+1}, y_t, \xi_{t+1}^y)\rangle + \frac{\ell}{2}\tau_2^2\|G_y(x_{t+1}, y_t, \xi_{t+1}^y)\|^2 \tag{16}$$
$$= \tau_2\left(1 + \frac{\ell}{2}\tau_2\right)\|\nabla_y f(x_{t+1}, y_t)\|^2 + \tau_2(1 + \ell\tau_2)\langle \nabla_y f(x_{t+1}, y_t), \Delta_t^y\rangle + \frac{\ell\tau_2^2}{2}\|\Delta_t^y\|^2,$$

where we used again the identity $G_y(x_{t+1}, y_t, \xi_{t+1}^y) = \nabla_y \hat{f}(x_{t+1}, y_t; z_t) + \Delta_t^y$.

Finally, we observe from the sm-AGDA update rule $z_{t+1} - z_t = \beta(x_{t+1} - z_t)$ that $\frac{1}{\beta}(z_{t+1} - z_t) = x_{t+1} - z_t$ and $(1 - \beta)(x_{t+1} - z_t) = (x_{t+1} - z_t) - (z_{t+1} - z_t) = x_{t+1} - z_{t+1}$. This gives

$$\hat{f}(x_{t+1}, y_{t+1}; z_{t+1}) - \hat{f}(x_{t+1}, y_{t+1}; z_t) = \frac{p}{2}\left[\|x_{t+1} - z_{t+1}\|^2 - \|x_{t+1} - z_t\|^2\right]$$
$$= \frac{p}{2}\left[\|(1-\beta)(x_{t+1} - z_t)\|^2 - \frac{1}{\beta^2}\|z_{t+1} - z_t\|^2\right]$$
$$= \frac{p}{2}\left[\frac{(1-\beta)^2}{\beta^2}\|z_{t+1} - z_t\|^2 - \frac{1}{\beta^2}\|z_{t+1} - z_t\|^2\right] \tag{17}$$
$$\leq \frac{-p}{2\beta}\|z_t - z_{t+1}\|^2,$$

where we used $0 < \beta \leq 1$. Therefore, summing up (15), (16), (17) yields the claim. $\square$

### B.2  Proof of Lemma 6

**Lemma 6.** *Suppose Assumptions 1, 2 and 3 hold, and $p \geq \ell$. Then, for any $t \in \mathbb{N}$,*

$$\Psi(y_{t+1}; z_{t+1}) - \Psi(y_t; z_t) \geq \tau_2\langle \nabla_y f(x^*(y_t, z_t), y_t), \ \nabla_y f(x_{t+1}, y_t)\rangle + \tau_2\langle \nabla_y f(x^*(y_t, z_t), y_t), \Delta_t^y\rangle$$

$$- \frac{L_\Psi}{2}\tau_2^2 \left(\|\nabla_y f(x_{t+1}, y_t)\|^2 + 2\langle \nabla_y f(x_{t+1}, y_t), \Delta_t^y\rangle + \|\Delta_t^y\|^2\right)$$

$$+ \frac{p}{2}\langle z_{t+1} - z_t, z_{t+1} + z_t - 2x^*(y_{t+1}, z_{t+1})\rangle,$$

where $L_\Psi \triangleq \ell\left(1 + \frac{p+\ell}{p-\ell}\right)$ and the map $x^*(\cdot, \cdot)$ is defined in (11).

*Proof.* By Lemma 19, $\Psi$ is $L_\Psi$-smooth in $y$ for any given $z \in \mathbb{R}^{d_1}$. Then, using $\nabla_y \Psi(y_t; z_t) = \nabla_y f(x^*(y_t, z_t), y_t)$, we obtain

$$
\begin{aligned}
\Psi(y_{t+1}, z_t) - \Psi(y_t, z_t) &\geq \langle \nabla_y f(x^*(y_t, z_t), y_t; z_t),\ y_{t+1} - y_t\rangle - \frac{L_\Psi}{2}\|y_{t+1} - y_t\|^2 \\
&\geq \tau_2 \langle \nabla_y f(x^*(y_t, z_t), y_t),\ \nabla_y f(x_{t+1}, y_t)\rangle + \tau_2 \langle \nabla_y f(x^*(y_t, z_t), y_t),\ \Delta_t^y\rangle \\
&\quad - \frac{L_\Psi}{2}\tau_2^2\left(\|\nabla_y f(x_{t+1}, y_t)\|^2 + 2\langle \nabla_y f(x_{t+1}, y_t), \Delta_t^y\rangle + \|\Delta_t^y\|^2\right).
\end{aligned}
$$
(18)

Furthermore, by definition of $\Psi$, we also have

$$
\begin{aligned}
\Psi(y_{t+1}; z_{t+1}) - \Psi(y_{t+1}; z_t) &= \hat{f}(x^*(y_{t+1}, z_{t+1}), y_{t+1}; z_{t+1}) - \hat{f}(x^*(y_{t+1}, z_t), y_{t+1}; z_t) \\
&\geq \hat{f}(x^*(y_{t+1}, z_{t+1}), y_{t+1}; z_{t+1}) - \hat{f}(x^*(y_{t+1}, z_{t+1}), y_{t+1}; z_t) \\
&= \frac{p}{2}\left[\|z_{t+1} - x^*(y_{t+1}, z_{t+1})\|^2 - \|z_t - x^*(y_{t+1}, z_{t+1})\|^2\right] \\
&= \frac{p}{2}(z_{t+1} - z_t)^\top [z_{t+1} + z_t - 2x^*(y_{t+1}, z_{t+1})].
\end{aligned}
$$
(19)

Summing (18) and (19), we conclude. $\square$

## C  Proof of Theorem 7

Before we move on to the proof of Theorem 7, we first provide a result from [70, 64] that quantifies the change in the function $P$ over the iterations. We also provide its proof for the sake of completeness.

**Lemma 15** (Lemma B.7 in [70]). *Suppose Assumptions 1, 2 and 3 hold. Consider the* `sm-AGDA` *iterate sequence* $(x_t, y_t, z_t)_{t\in\mathbb{N}}$ *for* $p > \ell$, *and let* $Y^*(z) \triangleq \arg\max_y \Psi(y; z)$ *denote the set of maximizers of* $\Psi(\cdot, z)$ *for given* $z$. *For any* $t \in \mathbb{N}$ *and* $y^*(z_{t+1}) \in Y^*(z_{t+1})$, *it holds that*

$$P(z_{t+1}) - P(z_t) \leq \frac{p}{2}\langle z_{t+1} - z_t,\ z_{t+1} + z_t - 2x^*(y^*(z_{t+1}), z_t)\rangle.$$

*Proof.* Let $y^*(z_{t+1}) \in Y^*(z_{t+1})$ and $y^*(z_t) \in Y^*(z_t)$ be two arbitrary maximizers. We have

$$
\begin{aligned}
P(z_{t+1}) - P(z_t) &= \min_x \max_y \hat{f}(x, y; z_{t+1}) - \min_x \max_y \hat{f}(x, y; z_t) \\
&= \max_y \min_x \hat{f}(x, y; z_{t+1}) - \max_y \min_x \hat{f}(x, y; z_t) \\
&= \Psi(y^*(z_{t+1}); z_{t+1}) - \Psi(y^*(z_t); z_t) \\
&\leq \Psi(y^*(z_{t+1}); z_{t+1}) - \Psi(y^*(z_{t+1}); z_t) \\
&= \hat{f}(x^*(y^*(z_{t+1}), z_{t+1}), y^*(z_{t+1}); z_{t+1}) - \hat{f}(x^*(y^*(z_{t+1}); z_t), y^*(z_{t+1}); z_t) \\
&\leq \hat{f}(x^*(y^*(z_{t+1}), z_t), y^*(z_{t+1}); z_{t+1}) - \hat{f}(x^*(y^*(z_{t+1}); z_t), y^*(z_{t+1}); z_t) \\
&= \frac{p}{2}(z_{t+1} - z_t)^\top [z_{t+1} + z_t - 2x^*(y^*(z_{t+1}), z_t)].
\end{aligned}
$$

The first and the third equality above hold by the definition of $P(z)$ and $\Psi(y; z)$ functions; on the other hand, for the second equality, one needs strong duality to hold. This interchange is valid and can be justified by the simple fact that $\hat{f}$ is strongly convex in $x$ (therefore, it satisfies the PL condition in $x$), and it also satisfies the PL condition in $y$ according to our assumption 2. It has been established in [65, Lemma 2.1] that the double-sided PL property allows one for the min-max switch. $\square$

Equipped with this lemma, the stage is set to prove Theorem 7 from Section 3. We first restate an extended version of this theorem, where the constants $\{c_i\}_{i=1}^8$ are provided explicitly.

**Theorem 7.** *Suppose Assumptions 1, 2, 3 and 4 hold. Consider the* `sm-AGDA` *algorithm with parameters* $p > \ell$, $\beta \in (0,1]$, $\tau_1 \in (0, \frac{1}{p+\ell}]$ *and* $\tau_2 > 0$ *chosen such that*

$$c_0 \triangleq -\tau_2^2 \ell \nu + \tau_2\left(1 - \frac{\ell}{2}\tau_2 - L_\Psi \tau_2\right) \geq 0, \quad c_0' \triangleq \frac{p}{3\beta} - \left(\frac{2p^2}{p-\ell} + 48\beta \frac{p^3}{(p-\ell)^2}\right) \geq 0$$

*for some constant* $\nu > 0$, *where* $L_\Psi = \ell\left(1 + \frac{p+\ell}{p-\ell}\right)$. *Then,*

$$\begin{aligned}
V_t - V_{t+1} \geq & c_1 \|\nabla_x \hat{f}(x_t, y_t; z_t)\|^2 + c_2 \|\nabla_y f(x^*(y_t, z_t), y_t)\|^2 + c_3 \|x_t - z_t\|^2 \\
& + c_4 \langle \nabla_x \hat{f}(x_t, y_t; z_t), \Delta_t^x \rangle + \langle c_5 \nabla_y f(x_t, y_t) + c_6 \nabla_y f(x^*(y_t, z_t), y_t), \Delta_t^y \rangle \\
& + c_7 \|\Delta_t^x\|^2 + c_8 \|\Delta_t^y\|^2,
\end{aligned} \tag{20}$$

*where the coefficients* $c_1$ *to* $c_8$ *have the following forms:*

$$c_1 = \frac{\tau_1}{2} - 2\left(\frac{1}{(p-\ell)^2} + \tau_1^2\right)\left(c_0 \ell^2 + \frac{\ell}{\nu} + 48\beta p\left(\frac{p+\ell}{p-\ell}\right)^2 \ell^2 \tau_2^2\right) - \left(c_0'\beta^2 + \frac{\ell}{6}(1 + \ell\tau_2 + 2L_\Psi\tau_2)\right)\tau_1^2,$$

$$c_2 = \frac{c_0}{2} - \frac{24\beta p}{(p-\ell)\mu}\left(1 + \tau_2 \frac{2p\ell}{p-\ell}\right)^2, \quad c_3 = c_0'\beta^2/2,$$

$$c_4 = -\left(192\beta p\left(\frac{p+\ell}{p-\ell}\right)^2 \ell^2 \tau_2^2 + \frac{4\ell}{\nu} + 4c_0\ell^2 + 2c_0'\beta^2\right)\tau_1^2 - \left((p+\ell)\tau_1 - 1\right)\tau_1,$$

$$c_5 = -\tau_2(1 + \ell\tau_2 + 2L_\Psi\tau_2), \quad c_6 = 2\tau_2,$$

$$c_7 = -\left(96\beta p\left(\frac{p+\ell}{p-\ell}\right)^2 \ell^2 \tau_2^2 + \frac{2\ell}{\nu} + \frac{p+\ell}{2} + 2c_0\ell^2 + \frac{\ell}{6}(1 + \ell\tau_2 + 2L_\Psi\tau_2) + c_0'\beta^2\right)\tau_1^2,$$

$$c_8 = -\left(48\beta p\left(\frac{p+\ell}{p-\ell}\right)^2 + \frac{\ell}{2} + L_\Psi + 3\ell(1 + \ell\tau_2 + 2L_\Psi\tau_2)\right)\tau_2^2.$$

*Furthermore, the* `sm-AGDA` *parameters can be chosen such that* $c_1, c_2, c_3 > 0$.

*Proof.* Combining the inequalities in Lemmas 5, 6 and 15 gives us a lower bound of the form:

$$V_t - V_{t+1} \geq A_1 + A_2 + A_3 + A_4 + A_5 + A_6, \tag{21}$$

for any $y^*(z_{t+1}) \in Y^*(z_{t+1})$ appearing in $A_3$, where

$$A_1 = \frac{\tau_1}{2}\|\nabla_x \hat{f}(x_t, y_t; z_t)\|^2 + \tau_2\left(1 - \frac{\ell}{2}\tau_2 - L_\Psi\tau_2\right)\|\nabla_y f(x_{t+1}, y_t)\|^2 + \frac{p}{2\beta}\|z_t - z_{t+1}\|^2,$$

$$A_2 = 2\tau_2 \langle \nabla_y f(x^*(y_t, z_t), y_t) - \nabla_y f(x_{t+1}, y_t), \nabla_y f(x_{t+1}, y_t)\rangle,$$

$$A_3 = 2p\langle z_{t+1} - z_t, x^*(y^*(z_{t+1}), z_t) - x^*(y_{t+1}, z_{t+1})\rangle$$

$$A_4 = -\tau_1((p+\ell)\tau_1 - 1)\langle \nabla_x \hat{f}(x_t, y_t; z_t), \Delta_t^x\rangle,$$

$$A_5 = \langle 2\tau_2 \nabla_y f(x^*(y_t, z_t), y_t) - \tau_2(1 + \ell\tau_2 + 2L_\Psi\tau_2)\nabla_y f(x_{t+1}, y_t), \Delta_t^y\rangle,$$

$$A_6 = -\frac{p+\ell}{2}\tau_1^2\|\Delta_t^x\|^2 - \frac{\ell\tau_2^2}{2}\|\Delta_t^y\|^2 - L_\Psi\tau_2^2\|\Delta_t^y\|^2.$$

Next, we provide lower bounds for several terms in the above inequality, including $A_2$, $A_3$, $A_5$, $\|\nabla_y f(x_{t+1}, y_t)\|^2$ and $\|z_{t+1} - z_t\|^2$. At the end, using these bounds within (21), we will be able to establish a descent property for $\{V_t\}$, which will allow to apply our concentration result (Theorem 9) for deriving the desired high probability bounds.

**Lower bound for** $A_2$. Using Cauchy-Schwarz inequality and $\ell$-smoothness of $f$, we have

$$\begin{aligned}
A_2 & \geq -2\tau_2 \|\nabla_y f(x^*(y_t, z_t), y_t) - \nabla_y f(x_{t+1}, y_t)\|\|\nabla_y f(x_{t+1}, y_t)\| \\
& \geq -2\tau_2 \ell \|x_{t+1} - x^*(y_t, z_t)\|\|\nabla_y f(x_{t+1}, y_t)\| \\
& \geq -\tau_2^2 \ell \nu \|\nabla_y f(x_{t+1}, y_t)\|^2 - \ell\nu^{-1}\|x_{t+1} - x^*(y_t, z_t)\|^2, \quad \forall \nu > 0,
\end{aligned}$$

where last line follows from Young's inequality. Thus, by Lemma 20, we obtain for any $\nu > 0$

$$A_2 \geq -\tau_2^2 \ell \nu \|\nabla_y f(x_{t+1}, y_t)\|^2 - \frac{2\ell}{\nu}\left(1 + \frac{1}{\tau_1^2(p-\ell)^2}\right)\tau_1^2 \|\nabla_x \hat{f}(x_t, y_t; z_t)\|^2$$
$$- \frac{4\ell}{\nu}\tau_1^2 \langle \nabla_x \hat{f}(x_t, y_t; z_t), \Delta_t^x \rangle - \frac{2\ell}{\nu}\tau_1^2 \|\Delta_t^x\|^2. \tag{22}$$

**Lower bound for $A_3$.** By Cauchy-Schwarz inequality and Lemma 21,

$$\begin{aligned}
A_3 &= 2p\langle z_{t+1} - z_t,\ x^*(y^*(z_{t+1}), z_t) - x^*(y^*(z_{t+1}), z_{t+1})\rangle \\
&\quad + 2p\langle z_{t+1} - z_t,\ x^*(y^*(z_{t+1}), z_{t+1}) - x^*(y_{t+1}, z_{t+1})\rangle \\
&\geq -2p\|z_{t+1} - z_t\| \|x^*(y^*(z_{t+1}), z_t) - x^*(y^*(z_{t+1}), z_{t+1})\| \\
&\quad + 2p\langle z_{t+1} - z_t,\ x^*(y^*(z_{t+1}), z_{t+1}) - x^*(y_{t+1}, z_{t+1})\rangle \\
&\geq -\frac{2p^2}{p-\ell}\|z_{t+1} - z_t\|^2 + 2p\langle z_{t+1} - z_t,\ x^*(y^*(z_{t+1}), z_{t+1}) - x^*(y_{t+1}, z_{t+1})\rangle.
\end{aligned}$$

Hence, using Young's inequality, for all $\beta > 0$, we obtain

$$A_3 \geq -\left(\frac{p}{6\beta} + \frac{2p^2}{p-\ell}\right)\|z_{t+1} - z_t\|^2 - 6\beta p \|x^*(y^*(z_{t+1}), z_{t+1}) - x^*(y_{t+1}, z_{t+1})\|^2. \tag{23}$$

We now lower bound the second term on the right-hand side of (23). First, note that we have $x^*(y^*(z_{t+1}), z_{t+1}) = x^*(z_{t+1})$, which follows from the fact that $\min_x \max_y \hat{f}(x, y; z) = \max_y \min_x \hat{f}(x, y; z)$ for all $z$ since $\hat{f}$ is strongly convex in $x$ and satisfies the PL condition in Assumption 2. Hence, using the inequality $\left\|\sum_{i=1}^N w_i\right\|^2 \leq N \sum_{i=1}^N \|w_i\|^2$, which holds for any $\{w_i\} \in \mathbb{R}^{d_1}$ and $N \geq 1$, we get

$$\begin{aligned}
&\|x^*(y^*(z_{t+1}), z_{t+1}) - x^*(y_{t+1}, z_{t+1})\|^2 \\
&\qquad \leq \ 4\|x^*(z_{t+1}) - x^*(z_t)\|^2 + 4\|x^*(z_t) - x^*(y_t^+(z_t), z_t)\|^2 \\
&\qquad\quad + 4\|x^*(y_t^+(z_t), z_t) - x^*(y_{t+1}, z_t)\|^2 + 4\|x^*(y_{t+1}, z_t) - x^*(y_{t+1}, z_{t+1})\|^2.
\end{aligned}$$

By Lemma 21 and Lemma 23, we observe that

$$4\|x^*(z_{t+1}) - x^*(z_t)\|^2 + 4\|x^*(y_{t+1}, z_t) - x^*(y_{t+1}, z_{t+1})\|^2 \leq \frac{8p^2}{(p-\ell)^2}\|z_{t+1} - z_t\|^2.$$

Using Lemma 24, we also have

$$4\|x^*(z_t) - x^*(y_t^+(z_t), z_t)\|^2 \leq \frac{4}{(p-\ell)\mu}\left(1 + \tau_2 \frac{2p\ell}{p-\ell}\right)^2 \|\nabla_y f(x^*(y_t, z_t), y_t)\|^2.$$

Finally, using Lemma 18, we get

$$\begin{aligned}
&4\|x^*(y_t^+(z_t), z_t) - x^*(y_{t+1}, z_t)\|^2 \\
&\qquad \leq 4\left(\frac{p+\ell}{p-\ell}\right)^2 \|y_t^+(z_t) - y_{t+1}\|^2 \\
&\qquad = 4\left(\frac{p+\ell}{p-\ell}\right)^2 \tau_2^2 \|\nabla_y f(x^*(y_t, z_t), y_t) - G_y(x_{t+1}, y_t, \xi_{t+1}^y)\|^2 \\
&\qquad = 4\left(\frac{p+\ell}{p-\ell}\right)^2 \tau_2^2 \|\nabla_y f(x^*(y_t, z_t), y_t) - (\nabla_y f(x_{t+1}, y_t) + \Delta_t^y)\|^2, \\
&\qquad \leq 4\left(\frac{p+\ell}{p-\ell}\right)^2 \tau_2^2 \left(2\ell^2 \|x^*(y_t, z_t) - x_{t+1}\|^2 + 2\|\Delta_t^y\|^2\right),
\end{aligned}$$

where the last inequality stems from the $\ell$-smoothness of $f$. In view of Lemma 20, we may further upper bound the above quantity as follows:

$$4\|x^*(y_t^+(z_t), z_t) - x^*(y_{t+1}, z_t)\|^2$$

$$\leq 16\Big(\frac{p+\ell}{p-\ell}\Big)^2 \ell^2 \tau_1^2 \tau_2^2 \Big[\Big(\frac{1}{\tau_1^2(p-\ell)^2}+1\Big)\|\nabla_x \hat{f}(x_t,y_t;z_t)\|^2 + 2\langle \nabla_x \hat{f}(x_t,y_t;z_t), \Delta_t^x\rangle + \|\Delta_t^x\|^2\Big]$$
$$+ 8\Big(\frac{p+\ell}{p-\ell}\Big)^2 \tau_2^2 \|\Delta_t^y\|^2.$$

Summing up the three intermediate inequalities we established above, we obtain

$$\|x^*(y^*(z_{t+1}), z_{t+1}) - x^*(y_{t+1}, z_{t+1})\|^2$$
$$\leq \frac{8p^2}{(p-\ell)^2}\|z_{t+1}-z_t\|^2 + \frac{4}{(p-\ell)\mu}\Big(1+\tau_2\ell + \frac{\tau_2\ell(p+\ell)}{p-\ell}\Big)^2 \|\nabla_y f(x^*(y_t,z_t),y_t)\|^2$$
$$+ 16\Big(\frac{p+\ell}{p-\ell}\Big)^2 \ell^2\Big(\frac{1}{\tau_1^2(p-\ell)^2}+1\Big)\tau_1^2\tau_2^2\|\nabla_x\hat{f}(x_t,y_t;z_t)\|^2$$
$$+ 32\Big(\frac{p+\ell}{p-\ell}\Big)^2\ell^2\tau_1^2\tau_2^2\langle\nabla_x\hat{f}(x_t,y_t;z_t),\Delta_t^x\rangle$$
$$+ 16\Big(\frac{p+\ell}{p-\ell}\Big)^2\ell^2\tau_1^2\tau_2^2\|\Delta_t^x\|^2 + 8\frac{(p+\ell)^2}{(p-\ell)^2}\tau_2^2\|\Delta_t^y\|^2.$$

In conclusion for $A_3$, using (23) and the above inequality, we obtain after some rearrangement

$$A_3 \geq -\Big(\frac{p}{6\beta} + \frac{2p^2}{p-\ell} + 48\beta\frac{p^3}{(p-\ell)^2}\Big)\|z_{t+1}-z_t\|^2 - \Big(96\beta p\Big(\frac{p+\ell}{p-\ell}\Big)^2\ell^2\tau_1^2\tau_2^2\Big)\|\Delta_t^x\|^2$$
$$- \Big(\frac{24\beta p}{(p-\ell)\mu}\Big(1+\tau_2\frac{2p\ell}{p-\ell}\Big)^2\Big)\|\nabla_y f(x^*(y_t,z_t),y_t)\|^2$$
$$- \Big[96\beta p\Big(\frac{p+\ell}{p-\ell}\Big)^2\ell^2\Big(\frac{1}{\tau_1^2(p-\ell)^2}+1\Big)\tau_1^2\tau_2^2\Big]\|\nabla_x\hat{f}(x_t,y_t;z_t)\|^2 \qquad (24)$$
$$- \Big(192\beta p\Big(\frac{p+\ell}{p-\ell}\Big)^2\ell^2\tau_1^2\tau_2^2\Big)\langle\nabla_x\hat{f}(x_t,y_t;z_t),\Delta_t^x\rangle - \Big(48\beta p\Big(\frac{p+\ell}{p-\ell}\Big)^2\tau_2^2\Big)\|\Delta_t^y\|^2.$$

**Lower bound for $A_5$.** Below we first bound $\langle\nabla_y f(x_{t+1},y_t),\Delta_t^y\rangle$, i.e.,

$$\langle\nabla_y f(x_{t+1},y_t),\Delta_t^y\rangle = \langle\nabla_y f(x_t,y_t),\Delta_t^y\rangle + \langle\nabla_y f(x_{t+1},y_t) - \nabla_y f(x_t,y_t),\Delta_t^y\rangle$$
$$\leq \langle\nabla_y f(x_t,y_t),\Delta_t^y\rangle + \frac{1}{12\tau_2\ell}\|\nabla_y f(x_{t+1},y_t)-\nabla_y f(x_t,y_t)\|^2 + 3\tau_2\ell\|\Delta_t^y\|^2$$
$$\leq \langle\nabla_y f(x_t,y_t),\Delta_t^y\rangle + \frac{\ell\tau_1^2}{12\tau_2}\|\hat{G}_x(x_t,y_t,\xi_{t+1}^x;z_t)\|^2 + 3\tau_2\ell\|\Delta_t^y\|^2$$
$$\leq \langle\nabla_y f(x_t,y_t),\Delta_t^y\rangle + \frac{\ell\tau_1^2}{6\tau_2}\|\nabla_x\hat{f}(x_t,y_t;z_t)\|^2 + \frac{\ell\tau_1^2}{6\tau_2}\|\Delta_t^x\|^2 + 3\tau_2\ell\|\Delta_t^y\|^2,$$

where the first inequality follows from Young's inequality, the second inequality follows from $\nabla_y f$ being $\ell$-smooth and the update rule $x_{t+1} = x_t - \tau_1\hat{G}_x(x_t,y_t,\xi_{t+1}^x;z_t)$, and in the third inequality we use $\Delta_t^x = \hat{G}_x(x_t,y_t,\xi_{t+1}^x;z_t) - \nabla_x\hat{f}(x_t,y_t;z_t)$. Thus, since $-\tau_2(1+\ell\tau_2+2L_\Psi\tau_2) < 0$, for any $\tau_2 > 0$, we obtain

$$A_5 \geq -\tau_2(1+\ell\tau_2+2L_\Psi\tau_2)\Big(\frac{\ell\tau_1^2}{6\tau_2}\|\nabla_x\hat{f}(x_t,y_t;z_t)\|^2 + \frac{\ell\tau_1^2}{6\tau_2}\|\Delta_t^x\|^2 + 3\tau_2\ell\|\Delta_t^y\|^2\Big) \qquad (25)$$
$$+ \langle 2\tau_2\nabla_y f(x^*(y_t,z_t),y_t) - \tau_2(1+\ell\tau_2+2L_\Psi\tau_2)\nabla_y f(x_t,y_t),\ \Delta_t^y\rangle.$$

**Lower bound for $\|\nabla_y f(x_{t+1},y_t)\|^2$.** Using Lemma 20 we can lower bound $\|\nabla_y f(x_{t+1},y_t)\|^2$ as follows:

$$\|\nabla_y f(x_{t+1},y_t)\|^2 \geq \frac{1}{2}\|\nabla_y f(x^*(y_t,z_t),y_t)\|^2 - \|\nabla_y f(x_{t+1},y_t) - \nabla_y f(x^*(y_t,z_t),y_t)\|^2$$
$$\geq \frac{1}{2}\|\nabla_y f(x^*(y_t,z_t),y_t)\|^2 - \ell^2\|x_{t+1}-x^*(y_t,z_t)\|^2$$
$$\geq \frac{1}{2}\|\nabla_y f(x^*(y_t,z_t),y_t)\|^2$$
$$- 2\ell^2\tau_1^2\Big[\Big(\frac{1}{\tau_1^2(p-\ell)^2}+1\Big)\|\nabla_x\hat{f}(x_t,y_t;z_t)\|^2 + 2\langle\nabla_x\hat{f}(x_t,y_t;z_t),\Delta_t^x\rangle + \|\Delta_t^x\|^2\Big].$$

We observe that $\|\nabla_y f(x_{t+1}, y_t)\|^2$ appears in both $A_1$ and the lower bound given in (22) for $A_2$; hence, grouping the two terms together and using the definition of $c_0 \geq 0$, we get

$$\left(\tau_2 \left(1 - \frac{\ell}{2}\tau_2 - L_\Psi \tau_2\right) - \tau_2^2 \ell \nu\right)\|\nabla_y f(x_{t+1}, y_t)\|^2 = c_0 \|\nabla_y f(x_{t+1}, y_t)\|^2$$

$$\geq \frac{c_0}{2}\|\nabla_y f(x^*(y_t, z_t), y_t)\|^2 - 2c_0 \ell^2 \left(\frac{1}{(p-\ell)^2} + \tau_1^2\right)\|\nabla_x \hat{f}(x_t, y_t; z_t)\|^2 \quad (26)$$

$$- 2c_0 \ell^2 \tau_1^2 \Big(2\langle \nabla_x \hat{f}(x_t, y_t; z_t), \Delta_t^x\rangle + \|\Delta_t^x\|^2\Big).$$

**Lower bound for $\|z_{t+1} - z_t\|^2$.** Since $z_{t+1} = z_t + \beta(x_{t+1} - z_t)$ for some $\beta > 0$, we observe that:

$$\|z_{t+1} - z_t\|^2 = \beta^2 \|x_{t+1} - z_t\|^2$$
$$= \beta^2 \|x_t - z_t - \tau_1 \hat{G}_x(x_t, y_t, \xi_{t+1}^x; z_t)\|^2$$
$$\geq \beta^2 \left(\frac{1}{2}\|x_t - z_t\|^2 - \tau_1^2 \|\hat{G}_x(x_t, y_t, \xi_{t+1}^x; z_t)\|^2\right),$$

and since $\hat{G}_x(x_t, y_t, \xi_{t+1}^x; z_t) = \nabla_x \hat{f}(x_t, y_t; z_t) + \Delta_t^x$, we obtain

$$\|z_t - z_{t+1}\|^2 \geq \frac{\beta^2}{2}\|x_t - z_t\|^2 - \beta^2 \tau_1^2 \|\nabla_x \hat{f}(x_t, y_t; z_t)\|^2 - 2\beta^2 \tau_1^2 \langle \nabla_x \hat{f}(x_t, y_t; z_t), \Delta_t^x\rangle - \beta^2 \tau_1^2 \|\Delta_t^x\|^2.$$

Note that $\|z_{t+1} - z_t\|^2$ appears in both $A_1$ and the lower bound given in (24) for $A_3$; hence, grouping the two terms together and using the definition of $c_0' \geq 0$ Using (27), we obtain

$$\left(\frac{p}{2\beta} - \frac{p}{6\beta} - \frac{2p^2}{p-\ell} - 48\beta \frac{p^3}{(p-\ell)^2}\right)\|z_{t+1} - z_t\|^2 = c_0' \|z_{t+1} - z_t\|^2$$

$$\geq -c_0'\beta^2 \tau_1^2 \|\nabla_x \hat{f}(x_t, y_t; z_t)\|^2 + c_0' \frac{\beta^2}{2}\|x_t - z_t\|^2 - 2c_0'\beta^2\tau_1^2\langle\nabla_x \hat{f}(x_t, y_t; z_t), \Delta_t^x\rangle - c_0'\beta^2\tau_1^2\|\Delta_t^x\|^2.$$

We conclude by combining all these lower bounds, i.e., the claimed Lyapunov descent inequality follows directly from summing (21), (22), (24), (25), (26) and (27). Finally, it follows after straightforward computations that there exist choice of sm-AGDA parameters which yield $c_1, c_2, c_3 > 0$ while satisfying the conditions $c_0 \geq 0$ and $c_0' \geq 0$; in fact Corollary 8 provides such sm-AGDA parameters explicitly. This completes the proof. $\qquad\square$

## D   Proof of Corollary 8

The lower bound provided in Theorem 7 resembles the descent property we require for our concentration result in Theorem 9. To allow for its proper application, we develop a stepsize policy inspired by [66].

**Corollary 8.** *Under the premise of Theorem 7, consider the parameters $p = 2\ell$, $\tau_1 \in (0, \frac{1}{3\ell}]$, $\tau_2 = \frac{\tau_1}{48}$, $\beta = \alpha\mu\tau_2$ for any $\alpha \in (0, \frac{1}{406}]$. Then, $\frac{\tilde{A}_{t+1} - \tilde{A}_t}{\tau_1} \leq -\tilde{B}_t + \tilde{C}_{t+1} + \tilde{D}_{t+1}$ for all $t \in \mathbb{N}$, where $\nu = \frac{12}{\tau_1\ell}$ and*

$$\tilde{A}_t \triangleq \tau_1 V_t, \quad \tilde{B}_t \triangleq \frac{\tau_1}{5}\|\nabla_x \hat{f}(x_t, y_t; z_t)\|^2 + \frac{\tau_2}{8}\|\nabla_y f(x^*(y_t, z_t), y_t)\|^2 + \frac{\beta p}{8}\|x_t - z_t\|^2$$

$$\tilde{C}_{t+1} \triangleq \left[\left(192\beta p\left(\frac{p+\ell}{p-\ell}\right)^2 \ell^2 \tau_2^2 + \frac{4\ell}{\nu} + 4c_0\ell^2 + 2c_0'\beta^2\right)\tau_1^2 + \left((p+\ell)\tau_1 - 1\right)\tau_1\right]\langle\nabla_x \hat{f}(x_t, y_t; z_t), \Delta_t^x\rangle$$

$$\quad + \tau_2\langle(1 + \ell\tau_2 + 2L_\Psi \tau_2)\nabla_y f(x_t, y_t) - 2\nabla_y f(x^*(y_t, z_t), y_t), \Delta_t^y\rangle,$$

$$\tilde{D}_{t+1} \triangleq 2\ell\tau_1^2\|\Delta_t^x\|^2 + 8\ell\tau_2^2\|\Delta_t^y\|^2.$$

*Proof.* In view of Theorem 7, it suffices to prove that setting $p = 2\ell$, $\tau_1 \in (0, \frac{1}{3\ell}]$, $\tau_2 = \frac{\tau_1}{48}$, $\beta = \alpha\mu\tau_2$ for for some positive $\alpha \leq \frac{1}{406}$ and $\nu = \frac{12}{\tau_1\ell}$ leads to both $c_0 \geq 0$ and $c_0' \geq 0$; furthermore, we also need to show that this choice of parameters implies the following lower bounds:

$$(i)\ c_1 \geq \tfrac{\tau_1}{5}, \quad (ii)\ c_2 \geq \tfrac{\tau_2}{8}, \quad (iii)\ c_3 \geq \tfrac{p\beta}{8}, \quad (iv)\ c_7 \geq -2\ell\tau_1^2, \quad (v)\ c_8 \geq -8\ell\tau_2^2.$$

First, we show that our parameter choice implies that $c_0, c_0' \geq 0$. Noting that $L_\Psi = 4\ell$, using $\tau_1 \leq \frac{1}{3\ell}$, we may bound $c_0$ from above and below as follows:

$$\tau_2 \geq c_0 \triangleq -\tau_2^2 \ell\nu + \tau_2\left(1 - \frac{\ell}{2}\tau_2 - L_\Psi\tau_2\right)$$

$$= \tau_2\left(1 - \left(\frac{\ell}{2} + L_\Psi + \ell\nu\right)\frac{\tau_1}{48}\right) \geq \tau_2\left(1 - \frac{9}{2\cdot 144} - \frac{1}{4}\right) \geq \frac{1}{2}\tau_2 \geq 0. \tag{27}$$

Moreover, since $\beta = \alpha\mu\tau_2$ and $\tau_2 \leq \frac{1}{144\ell}$, we get $\beta \leq \frac{\alpha}{144}$ using $\kappa \geq 1$. Hence, for $\alpha \leq \frac{1}{406}$,

$$0 \leq \frac{2p^2}{p-\ell} + 48\beta\frac{p^3}{(p-\ell)^2} = 8\ell(1+48\beta) = \left(\frac{\alpha}{36}(1+\alpha/3)\right)\frac{p}{\beta} \leq \frac{p}{12\beta}, \tag{28}$$

where the last inequality follows from $\alpha \leq \frac{1}{406} \leq \frac{3}{2}(\sqrt{5}-1)$; therefore, $c_0' \in [\frac{p}{4\beta}, \frac{p}{3\beta}]$.

Next, we prove bounds on $c_1, c_2, c_3, c_7, c_8$ separately.

**Proof of part $(i)$.** Since $p = 2\ell$, $\nu = \frac{12}{\tau_1\ell}$ and $\tau_1 \leq \frac{1}{3\ell}$, we first observe that

$$\frac{2\ell}{\nu}\left(\frac{1}{(p-\ell)^2} + \tau_1^2\right) = \frac{\tau_1}{6}(\tau_1^2\ell^2+1) \leq \frac{\tau_1}{6}\left(\frac{1}{9}+1\right) = \frac{5}{27}\tau_1. \tag{29}$$

Furthermore, using $\tau_2 = \frac{\tau_1}{48}$, and $\beta = \alpha\mu\tau_2$, we obtain

$$96\beta p\left(\frac{p+\ell}{p-\ell}\right)^2\ell^2\tau_2^2\left(\frac{1}{(p-\ell)^2} + \tau_1^2\right) = \left[96\beta\ell^2 p\frac{(p+\ell)^2}{(p-\ell)^4}\left(1 + \tau_1^2(p-\ell)^2\right)\frac{\tau_2^2}{\tau_1}\right]\tau_1$$

$$\leq \left[96\cdot\alpha\tau_2\mu\cdot 18\ell\left(1 + \frac{1}{9}\right)\frac{\tau_2^2}{\tau_1}\right]\tau_1$$

$$\leq 1920\cdot\alpha\cdot\frac{\tau_2^3\ell^2}{\tau_1}\cdot\frac{\mu}{\ell}\cdot\tau_1$$

$$\leq \frac{5\alpha}{2592}\tau_1, \tag{30}$$

where last line follows from the fact that $\mu/\ell \leq 1$ and $\frac{\tau_2^3\ell^2}{\tau_1} = \frac{\tau_1^2\ell^2}{48^3} \leq \frac{1}{144^2\cdot 48}$, in which we used $\tau_1 \leq \frac{1}{3\ell}$.

Since $\tau_2 \geq c_0 \geq 0$ and $-2\ell^2\left(\frac{1}{(p-\ell)^2}+\tau_1^2\right) \geq -\frac{20}{9}$, using $\tau_2 = \frac{\tau_1}{48}$, we obtain

$$-2c_0\ell^2\left(\frac{1}{(p-\ell)^2}+\tau_1^2\right) \geq -\frac{20\tau_2}{9} = -\frac{5}{108}\tau_1. \tag{31}$$

Using $L_\Psi = 4\ell$, $\tau_1 \leq \frac{1}{3\ell}$ and $\tau_2 \leq \frac{1}{144\ell}$, we get

$$\frac{\ell}{6}(1 + \ell\tau_2 + 2L_\Psi\tau_2)\tau_1^2 \leq \frac{\ell}{6}\left(1 + \frac{1}{144} + \frac{8}{144}\right)\tau_1^2 \leq \frac{1}{18}\left(1 + \frac{1}{144} + \frac{8}{144}\right)\tau_1 = \frac{17}{288}\tau_1. \tag{32}$$

Using $\tau_1 \leq \frac{1}{3\ell}$, $\tau_2 \leq \frac{1}{144\ell}$, this implies

$$-c_0'\tau_1^2\beta^2 \geq -\frac{p}{3}\beta\tau_1^2 \geq -\frac{2\ell}{3}\cdot\frac{\alpha\mu}{144\ell}\frac{1}{3\ell}\tau_1 \geq -\frac{\alpha}{648}\tau_1. \tag{33}$$

Combining equations (29), (30), (31), (32) and (33), we obtain

$$c_1 \geq \left(\frac{1}{2} - \frac{5}{27} - \frac{5\alpha}{2592} - \frac{5}{108} - \frac{17}{288} - \frac{\alpha}{648}\right)\tau_1 = \left(\frac{1}{2} - \frac{25}{108} - \frac{17+\alpha}{288}\right)\tau_1$$

$$= \frac{181-3\alpha}{864}\tau_1 \geq \frac{60-\alpha}{288} \geq \frac{\tau_1}{5}, \tag{34}$$

for $\alpha \le \frac{1}{406} < 2.4$, which completes the proof of part $(i)$.

**Proof of part $(ii)$.** Due to our parameter choice, we first note that

$$\frac{24\beta p}{(p-\ell)\mu}\left(1 + \tau_2\frac{2p\ell}{p-\ell}\right)^2 = 48\alpha\left(1 + \tau_2\frac{2p\ell}{p-\ell}\right)^2\tau_2 \le 96\alpha\tau_2 \le \frac{\tau_2}{8}, \tag{35}$$

where last line follows from $\tau_2 \le \frac{1}{144\ell}$ and $\alpha \le \frac{1}{406}$. Finally, (27) implies that $\frac{c_0}{2} \ge \frac{\tau_2}{4}$; hence, $c_2 \ge \left(\frac{1}{4} - \frac{1}{8}\right)\tau_2 = \frac{\tau_2}{8}$.

**Proof of part $(iii)$.** We have shown that $c_0' \ge \frac{p}{4\beta}$; hence, $c_3 \ge \frac{p\beta}{8}$.

**Proof of part $(iv)$.** According to (27), $\tau_2 \ge c_0$; hence, we deduce that

$$-2c_0\ell^2 \ge -2\tau_2\ell^2 \ge -\frac{1}{72}\ell, \tag{36}$$

where the last inequality follows from $\tau_2 \le \frac{1}{144\ell}$. Furthermore, we note that

$$-96\beta p\left(\frac{p+\ell}{p-\ell}\right)^2\ell^2\tau_2^2 = -1728\alpha\tau_2^3\ell^3\mu \ge -\frac{\alpha}{12^3}\ell, \tag{37}$$

with the last inequality following also from $\frac{\mu}{\ell} \le 1$ and $\tau_2 \le \frac{1}{144\ell}$. We also note that

$$-\frac{2\ell}{\nu} - \frac{p+\ell}{2} = -\left(\frac{\ell\tau_1}{6} + \frac{3}{2}\right)\ell \ge -\frac{14}{9}\ell.$$

Finally, since $c_0' \le \frac{p}{3\beta}$ according to (28), we get

$$-c_0'\beta^2 \ge -\beta^2\frac{p}{3\beta} = -\frac{2}{3}\alpha\mu\tau_2\ell \ge -\frac{\alpha}{216}\ell, \tag{38}$$

where we used $\tau_2 \le \frac{1}{144\ell}$. Thus, since $\alpha \in (0, 1)$, it follows from (36), (37), (38) and (32) that

$$c_7 \ge -\ell\tau_1^2\left(\frac{1}{72} + \frac{\alpha}{12^3} + \frac{14}{9} + \frac{\alpha}{216} + \frac{17}{96}\right) \ge -2\ell\tau_1^2.$$

**Proof of part $(v)$.** For $c_8$, we may simply observe that

$$\begin{aligned}
c_8 &= -\left(48\beta p\left(\frac{p+\ell}{p-\ell}\right)^2 + \frac{\ell}{2} + L_\Psi + 3\ell(1 + \ell\tau_2 + 2L_\Psi\tau_2)\right)\tau_2^2 \\
&= -\left(864\alpha\mu\tau_2 + \frac{1}{2} + 4 + 3\left(1 + \ell\tau_2 + 2L_\psi\tau_2\right)\right)\ell\tau_2^2 \\
&\ge -\left(6\alpha\frac{\mu}{\ell} + \frac{1}{2} + 4 + 3\left(1 + \frac{1}{144} + \frac{8}{144}\right)\right)\ell\tau_2^2 \ge -8\ell\tau_2^2,
\end{aligned}$$

where we used $\mu/\ell \le 1$, $L_\Psi = 4\ell$, $\tau_2 \le \frac{1}{144\ell}$, and $\alpha < \frac{1}{20}$. □

# E   Proof of Theorem 9

**Theorem 9.** *Let $\{\mathcal{F}_t\}_{t\in\mathbb{N}}$ be a filtration on $(\Omega, \mathcal{F}, \mathbb{P})$. Let $A_t, B_t, C_t, D_t$ be four stochastic processes adapted to the filtration such that there exist $\sigma_C, \sigma_D > 0$ and $\tau_1 > 0$ such that for all $t \in \mathbb{N}$: $(i)$ $B_t \ge 0$, $(ii)$ $\mathbb{E}[e^{\lambda C_{t+1}} \mid \mathcal{F}_t] \le e^{\lambda^2\sigma_C^2 B_t}$ for all $\lambda > 0$, $(iii)$ $\mathbb{E}[e^{\lambda D_{t+1}} \mid \mathcal{F}_t] \le e^{\lambda\sigma_D^2}$ for all $\lambda \in \left[0, \frac{1}{\sigma_D^2}\right]$ and $(iv)$ $\frac{A_{t+1}-A_t}{\tau_1} \le -B_t + C_{t+1} + D_{t+1}$. Then, for any $\bar{q} \in (0, 1]$, we have*

$$\mathbb{P}\left(\frac{\tau_1}{2}\sum_{t=0}^{T-1} B_t \le (A_0 - A_T) + \tau_1\sigma_D^2 T + 2\tau_1\max\{2\sigma_C^2, \sigma_D^2\}\log\left(\frac{1}{\bar{q}}\right)\right) \ge 1 - \bar{q}.$$

*Proof.* For some fixed $\gamma > 0$, let

$$S_T \triangleq \gamma \sum_{t=0}^{T-1} B_t - (A_0 - A_T)$$

with the convention that $S_0 \triangleq 0$. For any $T \in \mathbb{N}$, we have

$$
\begin{aligned}
S_{T+1} &= \gamma \sum_{t=0}^{T} B_t - (A_0 - A_{T+1}) \\
&= \gamma \sum_{t=0}^{T-1} B_t - (A_0 - A_T) + \gamma B_T - (A_T - A_{T+1}) \\
&= S_T + \gamma B_T - (A_T - A_{T+1}) \\
&= S_T + (\gamma - \tau_1) B_T + \tau_1 B_T - (A_T - A_{T+1}) \\
&\leq S_T + (\gamma - \tau_1) B_T + \tau_1 C_{T+1} + \tau_1 D_{T+1},
\end{aligned}
$$

where the inequality follows from condition *(iv)* of the hypothesis. Hence, for any $0 < \lambda \leq \frac{1}{2\tau_1 \sigma_D^2}$ and any $T \in \mathbb{N}$:

$$
\begin{aligned}
\mathbb{E}[e^{\lambda S_{T+1}} \mid \mathcal{F}_T] &\leq \mathbb{E}[e^{\lambda S_T} e^{\lambda(\gamma - \tau_1) B_T} e^{\lambda \tau_1 C_{T+1}} e^{\lambda \tau_1 D_{T+1}} \mid \mathcal{F}_T] \\
&\leq e^{\lambda S_T} e^{\lambda(\gamma - \tau_1) B_T} \mathbb{E}[e^{2\lambda \tau_1 C_{T+1}} \mid \mathcal{F}_T]^{\frac{1}{2}} \mathbb{E}[e^{2\lambda \tau_1 D_{T+1}} \mid \mathcal{F}_T]^{\frac{1}{2}} \\
&\leq e^{\lambda S_T} e^{\lambda(\gamma - \tau_1) B_T} \left( e^{4\lambda^2 \tau_1^2 \sigma_C^2 B_T} \right)^{\frac{1}{2}} \left( e^{2\lambda \tau_1 \sigma_D^2} \right)^{\frac{1}{2}} \\
&= e^{\lambda S_T} e^{\lambda(\gamma - \tau_1 + 2\tau_1^2 \lambda \sigma_C^2) B_T} e^{\lambda \tau_1 \sigma_D^2},
\end{aligned}
$$

where the first inequality follows from Cauchy-Schwarz and in the second one we use *(ii)* and *(iii)* of the hypothesis. Fixing $\gamma = \tau_1/2$ yields for all $0 < \lambda \leq \frac{1}{4\tau_1 \sigma_C^2}$:

$$\gamma - \tau_1 + 2\tau_1^2 \lambda \sigma_C^2 = \tau_1 \left( -\frac{1}{2} + 2\lambda \sigma_C^2 \tau_1 \right) \leq 0.$$

Therefore, for $0 < \lambda \leq \min \left\{ \frac{1}{4\tau_1 \sigma_C^2}, \frac{1}{2\tau_1 \sigma_D^2} \right\}$, using $B_T \geq 0$ by *(i)* of the hypothesis, we get

$$\mathbb{E}[e^{\lambda S_{T+1}} \mid \mathcal{F}_T] \leq e^{\lambda S_T} e^{\lambda \tau_1 \sigma_D^2},$$

and rolling this recursion backwards and noting $S_0 = 0$ yields:

$$\mathbb{E}[e^{\lambda S_T}] \leq e^{\lambda \tau_1 \sigma_D^2 T};$$

thus, using a Chernoff bound, we get

$$\mathbb{P}(S_T > t) \leq \mathbb{E}[e^{\lambda S_T}] e^{-\lambda t} \leq e^{\lambda(\tau_1 \sigma_D^2 T - t)}.$$

Since for $\bar{q} \in (0, 1]$,

$$e^{\lambda(\tau_1 \sigma_D^2 T - t)} \leq \bar{q} \iff t \geq \tau_1 \sigma_D^2 T - \frac{1}{\lambda} \log(\bar{q}),$$

we have

$$\mathbb{P}\left( \frac{\tau_1}{2} \sum_{t=0}^{T-1} B_t \leq (A_0 - A_T) + \tau_1 \sigma_D^2 T - \frac{1}{\lambda} \log(\bar{q}) \right) \geq 1 - \bar{q}.$$

The claim follows by taking $\lambda = \frac{1}{2\tau_1} \min \left\{ \frac{1}{2\sigma_C^2}, \frac{1}{\sigma_D^2} \right\}$. $\qquad\square$

# F Proof of Theorem 11

**Theorem 11.** *In the premise of Corollary 8,* `sm-AGDA` *iterates* $(x_t, y_t)$ *for* $\tau_1 \leq \frac{1}{3\ell}$ *satisfy*

$$\mathbb{P}\left(\frac{1}{T}\sum_{t=0}^{T-1}\left[\|\nabla_x f(x_t, y_t)\|^2 + \kappa\|\nabla_y f(x_t, y_t)\|^2\right] \leq \mathcal{Q}_{\bar{q},T},\right) \geq 1 - \bar{q}, \quad \forall\, T \in \mathbb{N}, \quad \forall\, \bar{q} \in (0, 1],$$

*for some* $\mathcal{Q}_{\bar{q},T} = \mathcal{O}\left(\frac{\kappa(\Delta_0 + b_0)}{\tau_1 T} + \kappa(\delta_x^2 + \delta_y^2)\left(\tau_1 \ell + \frac{1}{T}\log\left(\frac{1}{\bar{q}}\right)\right)\right)$ *explicitly stated in Appendix F,*
*where* $\Delta_0 \triangleq \Phi(z_0) - \Phi^*$, $b_0 \triangleq 2\sup_{x,y}\{\hat{f}(x_0, y; z_0) - \hat{f}(x, y_0; z_0)\}$.

As a first step, we provide a helper lemma that shows that our concentration result (Theorem 9) is applicable to the `sm-AGDA`-related processes $\tilde{A}_t, \tilde{B}_t, \tilde{C}_t, \tilde{D}_t$ introduced in Corollary 8.

**Lemma 16.** *Let* $A_t = \tilde{A}_t$, $B_t = \tilde{B}_t$, $C_t = \tilde{C}_t$, $D_t = \tilde{D}_t$, *where* $\tilde{A}_t, \tilde{B}_t, \tilde{C}_t, \tilde{D}_t$ *are defined in Corollary 8; moreover, let* $\tau_1 > 0$ *be the primal stepsize in* `sm-AGDA`*. Then, the processes* $A_t, B_t, C_t, D_t$ *are adapted to the filtration* $\mathcal{F}_t \triangleq \mathcal{F}_t^y$, *where* $\mathcal{F}_t^y$ *is defined in (6), and they satisfy the conditions of Theorem 9 with the following constants:*

$$(i)\ \ \sigma_C^2 = \tau_1(240\delta_x^2 + 32\delta_y^2), \quad (ii)\ \ \sigma_D^2 = 16\ell\tau_1^2\delta_x^2 + 64\ell\tau_2^2\delta_y^2.$$

*Proof.* The fact that $A_t, B_t, C_t, D_t$ are measurable with respect to $\mathcal{F}_t = \mathcal{F}_t^y$ for any $t \in \mathbb{N}$ follows directly from the definition of $\mathcal{F}_t$. Note that $x_t$ and $y_t$ are also $\mathcal{F}_t$-measurable for all $t \in \mathbb{N}$. We prove part $(i)$ and part $(ii)$ separately.

**Proof of part** $(i)$. First, recall that $C_{t+1} = \tilde{C}_{t+1} = -c_4\langle\nabla_x\hat{f}(x_t, y_t; z_t), \Delta_t^x\rangle - \langle c_5\nabla_y f(x_t, y_t) + c_6\nabla_y f(x^*(y_t, z_t), y_t), \Delta_t^y\rangle$. We would like to show that $\mathbb{E}[e^{\lambda C_{t+1}} \mid \mathcal{F}_t] \leq e^{\lambda^2\sigma_C^2 B_t}$ for all $\lambda > 0$. From Assumption 3, we note that for any $\lambda \geq 0$:

$$\mathbb{E}\left[\exp\left(\lambda\langle -c_4\nabla_x\hat{f}(x_t, y_t; z_t), \Delta_t^x\rangle\right) \mid \mathcal{F}_t\right] \leq \exp\left(8\lambda^2 c_4^2\|\nabla_x\hat{f}(x_t, y_t; z_t)\|^2\delta_x^2\right),$$

where we used [35, Lemma 3]. Now, given the value of $c_4$ in Theorem 7 and the convexity of $t \mapsto t^2$, we have

$$c_4^2 \leq 5\left(192\beta p\left(\frac{p+\ell}{p-\ell}\right)^2\ell^2\tau_2^2\tau_1^2\right)^2 + 5\tau_1^2\left((p+\ell)\tau_1 - 1\right)^2 + 5\left(\frac{4\ell}{\nu}\tau_1^2\right)^2 + 5\left(4c_0\ell^2\tau_1^2\right)^2 + 5\left(2c_0'\beta^2\tau_1^2\right)^2.$$

Now, leveraging the stepsize policy specified in Corollary 8, since $\alpha \in (0, 1)$, using (37) we get

$$5\left(192\beta p\left(\frac{p+\ell}{p-\ell}\right)^2\ell^2\tau_2^2\tau_1^2\right)^2 \leq 20 \cdot \left(\frac{\alpha}{12^3}\ell\tau_1^2\right)^2 \leq 20 \cdot \left(\frac{\alpha}{3 \cdot 12^3}\right)^2\tau_1^2 \leq \frac{\tau_1^2}{4}.$$

Similarly, as $\tau_1 \leq \frac{1}{3\ell}$, have $|(p+\ell)\tau_1 - 1)| \in [0, 1]$, which implies

$$5\tau_1^2\left((p+\ell)\tau_1 - 1\right)^2 \leq 5\tau_1^2.$$

Since $\nu = \frac{12}{\tau_1\ell}$, we have $5\left(\frac{4\ell}{\nu}\tau_1^2\right)^2 \leq \frac{5}{9}\ell^4\tau_1^6 \leq \frac{\tau_1^2}{4}$. Furthermore, using (27), we have $5\left(4c_0\ell^2\tau_1^2\right)^2 \leq 80\tau_1^4\tau_2^2\ell^4 \leq \frac{\tau_1^2}{4}$. Finally, (38) and $\alpha \in (0, 1)$ imply that

$$5\left(2c_0'\beta^2\tau_1^2\right)^2 \leq 20\left(\frac{\alpha}{216}\ell\tau_1\right)^2\tau_1^2 \leq \frac{\tau_1^2}{4}.$$

Thus, $c_4^2 \leq 6\tau_1^2$, which implies that

$$\mathbb{E}\left[\exp\left(-\lambda\langle c_4\nabla_x\hat{f}(x_t, y_t; z_t), \Delta_t^x\rangle\right) \mid \mathcal{F}_t\right] \leq \exp\left(48\lambda^2\tau_1^2\|\nabla_x\hat{f}(x_t, y_t; z_t)\|^2\delta_x^2\right). \quad (39)$$

Moreover, noting $\Delta_t^y$ is revealed after $\Delta_t^x$, using [35, Lemma 3] again along with the inequality $\|u + v\| \leq 2\|u\|^2 + 2\|v\|^2$, we have:

$$\mathbb{E}\left[\exp\left(-\lambda\langle c_5\nabla_y f(x_t, y_t) + c_6\nabla_y f(x^*(y_t, z_t), y_t), \Delta_t^y\rangle \mid \mathcal{F}_t, \Delta_t^x\right)\right) \quad (40)$$

$$= \mathbb{E}\left[\exp\left(\lambda\tau_2\langle(1 + \ell\tau_2 + 2L_\Psi\tau_2)\nabla_y f(x_t, y_t) - 2\nabla_y f(x^*(y_t, z_t), y_t), \Delta_t^y\rangle \mid \mathcal{F}_t, \Delta_t^x\right)\right]$$

$$\leq \exp\left(8\lambda^2\tau_2^2\|(1 + \ell\tau_2 + 2L_\Psi\tau_2)\nabla_y f(x_t, y_t) - 2\nabla_y f(x^*(y_t, z_t), y_t)\|^2\delta_y^2\right)$$

$$\leq \exp\left(64\lambda^2\tau_2^2\|\nabla_y f(x^*(y_t, z_t), y_t)\|^2\delta_y^2 + 16\lambda^2\tau_2^2(1 + \ell\tau_2 + 2L_\Psi\tau_2)^2\|\nabla_y f(x_t, y_t)\|^2\delta_y^2\right)$$

$$\leq \exp\left(64\lambda^2\tau_2^2\left(\|\nabla_y f(x^*(y_t, z_t), y_t)\|^2 + \|\nabla_y f(x_t, y_t)\|^2\right)\delta_y^2\right), \tag{41}$$

where the last line follows from $0 < 1 + \ell\tau_2 + 2L_\Psi\tau_2 \leq 1 + \frac{1}{144} + \frac{8}{144} \leq 2$.

Thus, since $\Delta_t^y$ is revealed after $\Delta_t^x$, using (39) and (41) together with the tower property of the conditional expectations, we get

$$\mathbb{E}\left[\exp\left(-\lambda\left(c_4\langle\nabla_x\hat{f}(x_t, y_t; z_t), \Delta_t^x\rangle + \langle c_5\nabla_y f(x_t, y_t) + c_6\nabla_y f(x^*(y_t, z_t), y_t), \Delta_t^y\rangle\right)\right)\right]$$

$$\leq \exp\left(48\lambda^2\tau_1^2\|\nabla_x\hat{f}(x_t, y_t; z_t)\|^2\delta_x^2 + 64\lambda^2\tau_2^2\left(\|\nabla_y f(x^*(y_t, z_t), y_t)\|^2 + \|\nabla_y f(x_t, y_t)\|^2\right)\delta_y^2\right). \tag{42}$$

Finally, observe that

$$\|\nabla_y f(x_t, y_t)\| - \|\nabla_y f(x^*(y_t, z_t), y_t)\| \leq \|\nabla_y f(x_t, y_t) - \nabla_y f(x^*(y_t, z_t), y_t)\|$$
$$\leq \ell\|x_t - x^*(y_t, z_t)\|$$
$$\leq \frac{\ell}{p - \ell}\|\nabla_x\hat{f}(x_t, y_t; z_t)\|,$$

where in the last inequality we used the $(p - \ell)$-strong convexity of $\hat{f}(\cdot, y_t; z_t)$ and the fact that we have $\nabla\hat{f}_x(x^*(y_t, z_t), y_t; z_t) = 0$. Therefore, since $p = 2\ell$, we obtain

$$\|\nabla_y f(x_t, y_t)\|^2 \leq 2\|\nabla_y f(x^*(y_t, z_t), y_t)\|^2 + 2\|\nabla_x\hat{f}(x_t, y_t; z_t)\|^2. \tag{43}$$

Plugging this inequality into (42) yields

$$\mathbb{E}\left[\exp\left(-\lambda\left(c_4\langle\nabla_x\hat{f}(x_t, y_t; z_t), \Delta_t^x\rangle + \langle c_5\nabla_y f(x_t, y_t) + c_6\nabla_y f(x^*(y_t, z_t), y_t), \Delta_t^y\rangle\right)\right)\right]$$

$$\leq \exp\left(\lambda^2\left(48\tau_1^2\delta_x^2 + 128\tau_2^2\delta_y^2\right)\|\nabla_x\hat{f}(x_t, y_t; z_t)\|^2 + 192\lambda^2\tau_2^2\delta_y^2\|\nabla_y f(x^*(y_t, z_t), y_t)\|^2\right)$$

$$\leq \exp\left(\lambda^2\max\left\{\tfrac{5}{\tau_1}\left(48\tau_1^2\delta_x^2 + 128\tau_2^2\delta_y^2\right), 1536\lambda^2\tau_2\delta_y^2\right\}B_t\right)$$

$$\leq \exp\left(\lambda^2\tau_1\left(240\delta_x^2 + 32\delta_y^2\right)B_t\right),$$

where the last inequality follows from $\tau_2 = \frac{\tau_1}{48}$.

**Proof of part** $(ii)$**.** Recall that $D_{t+1} \triangleq \tilde{D}_{t+1} = 2\ell\tau_1^2\|\Delta_t^x\|^2 + 8\ell\tau_2^2\|\Delta_t^y\|^2$. We would like to show that $\mathbb{E}[e^{\lambda D_{t+1}} \mid \mathcal{F}_t] \leq e^{\lambda\sigma_D^2}$ for all $\lambda \in \left[0, \frac{1}{\sigma_D^2}\right]$ for some $\sigma_D > 0$. First, observe that for any $\lambda > 0$ such that $\lambda \leq \min\{\frac{1}{16\ell\tau_1^2\delta_x^2}, \frac{1}{64\ell\tau_2^2\delta_y^2}\}$, we have

$$\mathbb{E}[e^{\lambda D_{t+1}} \mid \mathcal{F}_t] = \mathbb{E}\left[e^{2\lambda\ell\tau_1^2\|\Delta_t^x\|^2 + 8\lambda\ell\tau_2^2\|\Delta_t^y\|^2}\right]$$

$$\leq \mathbb{E}\left[e^{4\lambda\ell\tau_1^2\|\Delta_t^x\|^2}\right]^{\frac{1}{2}}\mathbb{E}\left[e^{16\lambda\ell\tau_2^2\|\Delta_t^y\|^2}\right]^{\frac{1}{2}}$$

$$\leq \left(e^{32\lambda\ell\tau_1^2\delta_x^2}\right)^{\frac{1}{2}}\left(e^{128\lambda\ell\tau_2^2\delta_y^2}\right)^{\frac{1}{2}}$$

$$\leq e^{\lambda(16\ell\tau_1^2\delta_x^2 + 64\ell\tau_2^2\delta_y^2)},$$

where we used [35, Lemma 2] in the second inequality. $\qquad\square$

Before completing the proof of Theorem 11, we provide another lemma that gives a lower bound on $B_t$ in terms of the squared norm of the partial gradients, based on our choice of `sm-AGDA` parameters.

**Lemma 17.** *For any $t \in \mathbb{N}$, we have*

$$\|\nabla_x f(x_t, y_t)\|^2 + \kappa \|\nabla_y f(x_t, y_t)\|^2 \leq \max\left\{\frac{20\kappa}{\tau_1}, \frac{16\kappa}{\tau_2}, \frac{32\ell}{\beta}\right\} B_t. \tag{44}$$

*Proof.* Since $\nabla_x \hat{f}(x_t, y_t; z_t) = \nabla_x f(x_t, y_t) + p(x_t - z_t)$, using $\|a + b\|^2 \leq 2\|a\|^2 + 2\|b\|^2$, we get

$$\|\nabla_x f(x_t, y_t)\|^2 \leq 2\|\nabla_x \hat{f}(x_t, y_t; z_t)\|^2 + 2p^2 \|x_t - z_t\|^2.$$

Hence, together with (43) and the definition of $B_t$, we obtain

$$\|\nabla_x f(x_t, y_t)\|^2 + \kappa \|\nabla_y f(x_t, y_t)\|^2$$
$$\leq (2 + 2\kappa)\|\nabla_x \hat{f}(x_t, y_t; z_t)\|^2 + 2\kappa \|\nabla_y f(x^*(y_t, z_t), y_t)\|^2 + 2p^2 \|x_t - z_t\|^2$$
$$\leq \max\left\{\frac{(2 + 2\kappa) \cdot 5}{\tau_1}, \frac{2\kappa \cdot 8}{\tau_2}, \frac{2p^2 \cdot 8}{\beta p}\right\} B_t$$
$$\leq \max\left\{\frac{20\kappa}{\tau_1}, \frac{16\kappa}{\tau_2}, \frac{32\ell}{\beta}\right\} B_t.$$

with the last inequality following from $\kappa \geq 1$. $\square$

We may now provide our proof for Theorem 11.

*Proof of Theorem 11.* According to Lemma 16, the processes $A_t, B_t, C_t$, and $D_t$ defined in the statement of Lemma 16 satisfy the conditions of Theorem 9 if we set $\tau_1$ as the primal stepsize of `sm-AGDA`. Therefore, for any $\bar{q} \in [0, 1)$,

$$\mathbb{P}\left(\frac{\tau_1}{2} \sum_{t=0}^{T-1} B_t \leq \tau_1(V_0 - V_T) + \tau_1 \sigma_D^2 T + 2\tau_1 \max\left\{2\sigma_C^2, \sigma_D^2\right\} \log\left(\frac{1}{\bar{q}}\right)\right) \geq 1 - \bar{q}.$$

Thus, dividing by $\frac{\tau_1}{2}T$ and using Lemma 17, we can conclude that with the probability at least $1 - \bar{q}$, the following event holds:

$$\frac{1}{T} \sum_{t=0}^{T-1} \|\nabla_x f(x_t, y_t)\|^2 + \kappa \|\nabla_y f(x_t, y_t)\|^2$$
$$\leq 2 \max\left\{\frac{20\kappa}{\tau_1}, \frac{16\kappa}{\tau_2}, \frac{32\ell}{\beta}\right\} \left(\frac{1}{T}(V_0 - V_T) + \sigma_D^2 + \frac{1}{T} \max\left\{4\sigma_C^2, 2\sigma_D^2\right\} \log\left(\frac{1}{\bar{q}}\right)\right). \tag{45}$$

Finally, we can relate the potential gap $V_0 - V_T$ to the primal suboptimality, i.e., $\Delta_0$, and to the duality gap, i.e., $b_0/2 = \sup_{x,y}\{\hat{f}(x_0, y; z_0) - \hat{f}(x, y_0; z_0)\}$, at the initialization, following the same arguments provided in [66]. More precisely, for $V(x, y, z) \triangleq \hat{f}(x, y; z) - 2\Psi(y, z) + 2P(z)$, we first observe that

$$V_0 - V_T \leq V_0 - \min_{x,y,z} V(x, y, z) = V_0 - \min_{x,y,z} \hat{f}(x, y; z) - 2\Psi(y, z) + 2P(z)$$
$$\leq V_0 - \min_z P(z). \tag{46}$$

where last line follows from $\hat{f}(x, y; z) - \Psi(y, z) \geq 0$ for all $y, z$ and $P(z) - \Psi(y, z) \geq 0$ for all $y, z$. Since $p = 2\ell$, we also note that

$$P(z_0) = \min_x \max_y f(x, y) + \ell\|x - z_0\|^2 \leq \max_y f(z_0, y) = \Phi(z_0).$$

Finally, since $P$ is the Moreau envelope of $\Phi$, we have $\min_z P(z) = \min_z \Phi(z)$; therefore,

$$V_0 - \min_z P(z) = \hat{f}(x_0, y_0; z_0) - 2\Psi(y_0, z_0) + 2P(z_0) - \min_z \Phi(z)$$
$$\leq \Phi(z_0) - \min_z \Phi(z) + \hat{f}(x_0, y_0; z_0) - \Psi(y_0, z_0) + P(z_0) - \Psi(y_0, z_0) \tag{47}$$
$$\leq \Phi(z_0) - \min_z \Phi(z) + \frac{1}{2}b_0 + \frac{1}{2}b_0 = \Delta_0 + b_0.$$

Note that $\tau_2 = \frac{\tau_1}{48}$ implies $16\kappa/\tau_2 \geq 20\kappa/\tau_1$, and $32\ell/\beta = \frac{32}{\alpha} \cdot \kappa/\tau_2 > 16\kappa/\tau_2$ since $\alpha \in (0,1)$. Therefore,

$$2 \max \left\{ \frac{20\kappa}{\tau_1}, \frac{16\kappa}{\tau_2}, \frac{32\ell}{\beta} \right\} = \frac{64}{\alpha} \cdot \kappa/\tau_2. \tag{48}$$

Therefore, combining (45), (46), (47) and (48), we conclude that $\mathcal{Q}_{q,T}$ has the following explicit form:

$$\mathcal{Q}_{\bar{q},T} = r_1 \left\{ \frac{\Delta_0 + b_0}{T} + r_2 + \frac{r_3}{T} \log\left(\frac{1}{\bar{q}}\right) \right\}, \tag{49}$$

where the constants $r_1, r_2$ and $r_3$ are defined as

$$r_1 = \frac{64}{\alpha} \frac{\kappa}{\tau_2}, \ r_2 = \sigma_D^2 = 16\ell\tau_1 \left( \tau_1 \delta_x^2 + \frac{1}{12}\tau_2\delta_y^2 \right), \ r_3 = \max\{4\sigma_C^2, 2\sigma_D^2\} = 4\sigma_C^2 = 4\tau_1(240\delta_x^2 + 32\delta_y^2),$$

where the equalities follow from the expressions of $\sigma_C^2$ and $\sigma_D^2$ provided in Lemma 16. $\qquad\square$

# G  Proof of Corollary 14

Setting $\tau_2 = \tau_1/48$ for any $\tau_1 > 0$ implies that $\mathcal{Q}_{\bar{q},T}$ defined in (49) satisfies $\mathcal{Q}_{\bar{q},T} = \mathcal{O}\left( \frac{\kappa(\Delta_0+b)}{\tau_1 T} + \tau_1 \ell\kappa\delta^2 + \frac{\delta^2 \kappa}{T} \log\left(\frac{1}{\bar{q}}\right) \right)$. Hence, setting $\tau_1 = \min\left( \frac{1}{3\ell}, \frac{48\sqrt{\Delta_0+b_0}}{\sqrt{T\ell\delta^2}} \right)$ ensures that

$$\mathcal{Q}_{q,T} = \mathcal{O}\left( \frac{(\Delta_0 + b_0)\ell\kappa}{T} + \sqrt{\frac{(\Delta_0 + b_0)\ell}{T}}\delta\kappa + \frac{\delta^2 \kappa}{T} \log\left(\frac{1}{\bar{q}}\right) \right).$$

Finally, to obtain an $\mathcal{O}(\varepsilon, \varepsilon/\sqrt{\kappa})$-stationary point, it suffices to have the above bound smaller than $\mathcal{O}(\varepsilon^2)$, which is guaranteed when the following three conditions are met up to a constant factor: *(i)* $\frac{(\Delta_0+b_0)\ell\kappa}{T} \leq \frac{\varepsilon^2}{3}$, *(ii)* $\sqrt{\frac{(\Delta_0+b_0)\ell}{T}}\delta\kappa \leq \frac{\varepsilon^2}{3}$, and *(iii)* $\frac{\delta^2\kappa}{T} \log\left(\frac{1}{\bar{q}}\right) \leq \frac{\varepsilon^2}{3}$. This is directly implied by the value $T_{\varepsilon,\bar{q}}$ given in our corollary statement.

# H  Supplementary Lemmas

In this section, we present a sequence of supplementary lemmas essential for deriving our main result, Theorem (11). Some of these results are well-known and are directly referenced. For others, we have improved specific algebraic constants. Additionally, some lemmas are extensions of existing bounds provided in expectation.

**Lemma 18.** *For any $z \in \mathbb{R}^{d_1}$ and $y_1, y_2 \in \mathbb{R}^{d_2}$, $\|x^*(y_1, z) - x^*(y_2, z)\| \leq \left(\frac{p+\ell}{p-\ell}\right)\|y_1 - y_2\|$.*

*Proof.* This result is provided in [66, Lemma C.1], which immediately follows from [41, Lemma B.2, part (c)]. $\qquad\square$

**Lemma 19.** *For any $z \in \mathbb{R}^{d_1}$, the map $y \mapsto \Psi(y,z) = \min_x \hat{f}(x,y,z)$ is $\ell\left(1 + \frac{p+\ell}{p-\ell}\right)$-smooth.*

*Proof.* This result immediately follows from [41, Lemma B.2, part (d)]. Indeed, for any $y_1, y_2 \in \mathbb{R}^{d_2}$, we have

$$\|\nabla_y \Psi(y_1, z) - \nabla_y \Psi(y_2, z)\|$$
$$= \|\nabla_y f(x^*(y_1, z), y_1) - \nabla_y f(x^*(y_2, z), y_2)\|$$
$$\leq \ell\|x^*(y_1, z) - x^*(y_2, z)\| + \ell\|y_1 - y_2\| \leq \left(1 + \frac{p+\ell}{p-\ell}\right)\ell\|y_1 - y_2\|,$$

where the first equality follows from Danskin's theorem, the first inequality follos from $\ell$-smoothness of $f$, and the last inequality follows from Lemma 18. $\qquad\square$

**Lemma 20.** *Consider the iterates $(x_t, y_t, z_t)$ of the* sm-AGDA *algorithm. For any $t \in \mathbb{N}$,*

$$\begin{aligned} \|x_{t+1} - x^*(y_t, z_t)\|^2 &\leq 2\left(1 + \frac{1}{\tau_1^2(p-\ell)^2}\right)\tau_1^2\|\nabla_x \hat{f}(x_t, y_t; z_t)\|^2 \\ &\quad + 4\tau_1^2 \langle \nabla_x \hat{f}(x_t, y_t; z_t), \Delta_t^x \rangle + 2\tau_1^2\|\Delta_t^x\|^2. \end{aligned}$$

*Proof.* For any $t \in \mathbb{N}$, using the fact that $\hat{f}$ is strongly convex in $x$ with modulus $(p - \ell)$ together with $x^*(y_t, z_t) = \operatorname{argmin}_x \hat{f}(x, y_t; z_t)$, and $x_{t+1} = x_t - \tau_1 \hat{G}_x(x_t, y_t, \xi_{t+1}^x; z_t)$, we get

$$\|x_{t+1} - x^*(y_t, z_t)\|^2 \leq 2\|x_t - x^*(y_t, z_t)\|^2 + 2\|x_{t+1} - x_t\|^2$$

$$\leq \frac{2}{(p-\ell)^2}\|\nabla_x \hat{f}(x_t, y_t; z_t)\|^2 + 2\tau_1^2 \|\hat{G}_x(x_t, y_t, \xi_{t+1}^x; z_t)\|^2.$$

Using the identity $\hat{G}_x(x_t, y_t, \xi_{t+1}^x; z_t) = \Delta_t^x + \nabla_x \hat{f}(x_t, y_t; z_t)$ within the above inequality yields the desired result. $\qquad\square$

**Lemma 21.** *For any $y \in \mathbb{R}^{d_2}$, $z_1, z_2 \in \mathbb{R}^{d_1}$, we have:*

$$\|x^*(y, z_1) - x^*(y, z_2)\| \leq \frac{p}{p - \ell}\|z_1 - z_2\|.$$

*Proof.* This result follows from [70, Lemma B.2]. Indeed, since $\hat{f}$ is strongly convex in $x$ with modulus $p - \ell$, we get

$$\hat{f}(x^*(y, z_1), y; z_2) - \hat{f}(x^*(y, z_2), y; z_2) \geq \frac{p - \ell}{2}\|x^*(y, z_1) - x^*(y, z_2)\|^2. \tag{50}$$

Then swapping $z_1, z_2$ leads to

$$\hat{f}(x^*(y, z_2), y; z_1) - \hat{f}(x^*(y, z_1), y; z_1) \geq \frac{p - \ell}{2}\|x^*(y, z_1) - x^*(y, z_2)\|^2. \tag{51}$$

Furthermore, from the definition $\hat{f}$, it follows that

$$\hat{f}(x^*(y, z_2), y; z_1) - \hat{f}(x^*(y, z_2), y; z_2) = \frac{p}{2}\left(\|x^*(y, z_2) - z_1\|^2 - \|x^*(y, z_2) - z_2\|^2\right)$$

$$= \frac{p}{2}\left(\|z_1\|^2 - \|z_2\|^2 + 2\langle x^*(y, z_2), z_2 - z_1\rangle\right);$$

similarly, swapping $z_1, z_2$ in the above inequality leads to

$$\hat{f}(x^*(y, z_1), y; z_2) - \hat{f}(x^*(y, z_1), y; z_1) = \frac{p}{2}\left(\|x^*(y, z_1) - z_2\|^2 - \|x^*(y, z_1) - z_1\|^2\right)$$

$$= \frac{p}{2}\left(\|z_2\|^2 - \|z_1\|^2 - 2\langle x^*(y, z_1), z_2 - z_1\rangle\right).$$

Thus, summing the above two identities and applying Cauchy-Schwartz gives us

$$\hat{f}(x^*(y, z_2), y; z_1) - \hat{f}(x^*(y, z_2), y; z_2) + \hat{f}(x^*(y, z_1), y; z_2) - \hat{f}(x^*(y, z_1), y; z_1)$$

$$\leq p\|x^*(y, z_1) - x^*(y, z_2)\|\|z_1 - z_2\|.$$

We can lower bound the left hand side of the above inequality using (50) and (51), which leads to

$$(p - \ell)\|x^*(y, z_1) - x^*(y, z_2)\|^2 \leq p\|x^*(y, z_1) - x^*(y, z_2)\|\|z_1 - z_2\|. \tag{52}$$

Rearranging this inequality yields the desired result. $\qquad\square$

**Lemma 22.** *For any $x \in \mathbb{R}^{d_1}$, $x \mapsto \Phi(x; z)$ is $(p - \ell)$-strongly convex.*

*Proof.* For any $y \in \mathbb{R}^{d_2}, z \in \mathbb{R}^{d_1}$, $x \mapsto \hat{f}(x, y; z)$ is strongly convex with modulus $p - \ell$, i.e., $x \mapsto \hat{f}(x, y; z) - \frac{p-\ell}{2}\|x\|^2$ is convex. Then, $x \mapsto \sup_{y \in \mathbb{R}^{d_2}} \hat{f}(x, y; z) - \frac{p-\ell}{2}\|x\|^2$ is convex as it is the pointwise supremum of convex functions. Therefore, $x \mapsto \Phi(x; z) - \frac{p-\ell}{2}\|x\|^2$ is convex, and this implies $x \mapsto \Phi(x, z)$ is strongly convex with modulus $p - \ell$. $\qquad\square$

**Lemma 23.** *For all $z_1, z_2 \in \mathbb{R}^{d_1}$, we have*

$$\|x^*(z_1) - x^*(z_2)\| \leq \frac{p}{p - \ell}\|z_1 - z_2\|.$$

*Proof.* Using the result of Lemma 22, one can show this result following exactly the same arguments in the proof of Lemma 21. $\qquad\square$

**Lemma 24.** *For any $y \in \mathbb{R}^{d_2}$, $z \in \mathbb{R}^{d_1}$, it holds that*

$$\|x^*(z) - x^*(y^+(z), z)\|^2 \leq \frac{1}{(p - \ell)\mu}\left(1 + \tau_2 \ell \frac{2p}{p - \ell}\right)^2 \|\nabla_y f(x^*(y, z), y)\|^2.$$

*Proof.* The proof is the same as [66, Lemma C.2]. $\qquad\square$

# I   Further Details about Figure 2

In this section, we provide further details about how the empirical and theoretical quantiles are estimated in Figure 2 and are compared.

**Generation of the theoretical quantiles $\mathcal{Q}_{q,T}$.**   For any given $q \in (0,1)$, estimating an upper bound on $\mathcal{Q}_{q,T}$ requires estimating an upper bound on the quantity $\Delta_0 + b_0$ based on Theorem 11. Other constants such as $r_1, r_2, r_3$, $\ell$, and $\mu$ are explicitly known in the setting of this experiment based on the NCPL game where $T = 10{,}000$ is fixed.

First, we set the initial point $(x_0, y_0)$ randomly, where each component of $x_0$ and $y_0$ is sampled uniformly from the interval $[-20, 20]$, and we set $z_0 = x_0$. We then estimate an upper bound on the quantity $\Delta_0 + b_0$ numerically based on a grid search, resulting in $\Delta_0 + b_0 = 12$. Second, we generate a linear mesh $I_m$ with a grid size $m = 0.0002$ over the interval $[0, 1]$. For $q \in I_m$, we calculate $\mathcal{Q}_{q,T}$ based on Theorem 11. Third, we generate a sequence of quantiles $\mathcal{Q}_{I_m,T}$. These quantiles are used to create a CDF via linear interpolation using the `scipy.interpolate` package's `interp1d` function in Python. Note that this quantile sequence generates a CDF over the values $\mathcal{Q}_{I_m,T}$.

**Generation of the empirical quantiles of the random variable $X_T$.**   We generate 1,000 samples $\{X_T^{(i)}\}_{i=1}^{1000}$ from the sample paths corresponding to the NCPL game with $T = 10{,}000$, where $X_T = \frac{1}{T}\sum_{t=1}^{T} \|\nabla_x f(x_t, y_t)\|^2 + \kappa \|\nabla_y f(x_t, y_t)\|^2$ represents the path averages of the gradients. Quantiles for this sequence were generated over $I_m$ using NumPy's quantile generator in Python, ensuring alignment with the mesh over which the theoretical quantiles were generated. Evidently, our theoretical quantiles dominate the empirical quantiles pointwise, demonstrating that in the challenging NCPL regime, our theory provides empirically verifiable guarantees on the tail behavior of the random variable $X_T$.

**Comparison of quantiles.**   We plotted the CDF corresponding to the theoretical quantiles $\mathcal{Q}_{q,T}$ over the values of the empirical quantiles using a common mesh grid over the range of the empirical averages of the sample paths. In other words, we scaled the quantiles $\mathcal{Q}_{q,T}$ with an affine transformation so that their range matches the range of the empirical quantiles. This affine scaling preserves the shape of the distribution corresponding to the theoretical quantiles and allows for better visualization.

