# OpenReview forum: "High-probability complexity bounds for stochastic non-convex minimax optimization"
_NeurIPS.cc/2024/Conference — NeurIPS 2024 poster_

### Official Review · Reviewer_pQzv · 2024-07-12

**Soundness:** 3
**Presentation:** 3
**Contribution:** 3
**Rating:** 7
**Confidence:** 4

**Summary:**

This paper proves a high probability convergence bound and almost sure convergence for stochastic smoothed AGDA method for minimax problems in nonconvex-PL setting. This is the first high probability guarantee and almost sure convergence guarantee in this nonconvex setting.

**Strengths:**

This is the first high probability convergence guarantee and almost sure convergence guarantee in this nonconvex setting, going beyond convex setting and convergence in expectation.
The paper also presents a concentration bound (Theorem 9) which is of independent interests and this bound allows us to obtain high probability convergence results from expectation convergence results for minimax problems.

**Weaknesses:**

See Questions.

**Questions:**

1. The algorithm and analysis are built upon the approaches in 63 and 68. Could the authors provide more insights on the novelty of the proof compared to 63 and 68?
2. Is it possible to extend this analysis to other settings like the nonconvex-concave setting?
3. Can this analysis be extended to other algorithms such as stochastic GDA?
4. Missing references:
[1]. Shen et al   Stochastic gradient descent ascent for federated minimax optimization
[2]  Zheng et al Universal gradient descent ascent method for nonconvex-nonconcave minimax optimization
[3].  Ozdaglar et al What is a good metric to study the generalization of minimax learners
[4] Li et al Nonsmooth-nonconvex-nonconcave minimax optimization: primal-dual balancing and iteration complexity analysis

---

> ### Author Rebuttal · Authors · 2024-08-05
>
> Thanks for all the insightful comments. Below are our responses:
>
> **1-** [68] analyzes sm-AGDA in the deterministic nonconvex-concave setting, and introduces a Lyapunov function to establish a complexity result. In a follow-up work, [63] leverages the same Lyapunov function to adapt the analysis to the stochastic nonconvex-PL setting, and establishes $O(\ell\kappa / \epsilon^2 + \ell\kappa^2 \sigma^2 / \epsilon^4)$ complexity; $\kappa$ is the condition number, and $\sigma^2$ bounds the variance.
>
> Some  Lyapunov arguments in our proof are built upon these analyses to establish our high-probability (HP) bound. In fact, it is a common strategy in the literature on HP bounds to build upon existing deterministic/expectation analyses [81,82]. Yet the key difficulty remains in controlling how the noise from gradients accumulates along iterations. While one may intuitively expect that the gradient estimates being subGaussian should produce a light-tail distribution for sm-AGDA’s output, achieving a precise balance between bias and quantile trade-off in the complexity bound usually turns out to be highly nontrivial – even in the strongly convex-strongly concave (SCSC) setting [34].
>
> In the nonconvex-PL case, we show that for finding an $\epsilon$-solution w.p. at least $1 - q$, the complexity of sm-AGDA is $O(\ell\kappa / \epsilon^2 + \delta^2 \log(1/q) \kappa / \epsilon^2 + \ell\kappa^2 \delta^2 / \epsilon^4)$ under the subGaussian assumption on the gradients. This bound is tight: for the deterministic case, i.e., when $\delta = 0$, we recover the existing $O(\kappa / \epsilon^2)$ complexity bound for sm-AGDA; and the quantile dependence of our bounds scales favorably with $\epsilon$ and $q$. Our bound is tighter than those obtained by the standard expectation to HP conversions, as we explain in our general response to the reviewers above.
>
> Our main technical novelty behind this result is the identification of stochastic processes (see $ \tilde{A}_t, \tilde{B}_t, \tilde{C}_t,$ and $\tilde{D}_t$ in Cor. 8) that exhibit the desired concentration properties as stated in our generic concentration result (Thm. 9). Constructing these processes, e.g., $\tilde{C}_t$, is non-trivial, requiring careful measurability considerations along with deriving some new convex inequalities to extend the existing analysis for deriving expectation guarantees [63] and for the deterministic setting [68] of sm-AGDA. These challenges are specific to HP analyses and do not occur in the less sensitive expectation bounds from [63], where noise can be mitigated using the properties of conditional expectation. For a discussion on the challenges related to measurability in the Lyapunov analysis of momentum-averaged algorithms, we refer to [34], which obtained HP bounds in the SCSC setting. Similarly, sm-AGDA employs momentum averaging in the non-convex setting whenever $\beta > 0$, which complicates the analysis with measurability issues comparable to those arising in [34].
>
> Finally, a surprising result of our analysis, contrary to naive approaches leveraging Markov’s inequality by running the method several times in parallel, is that the cost of getting a HP bound only affects negligible terms of the initial expectation bounds giving $ O(\epsilon^{-4} + \log(1/q) \epsilon^{-2}) $ complexity, instead of multiplying the whole bound by $\log(1/q)$ to get a worse bound of $O(\log(1/q) \epsilon^{-4})$ - see our general response in that regard. This feature departs from standard guarantees in the SCSC setting [34].
>
> **2-** In our proof, we use the PL property of the dual to achieve an appropriate Lyapunov descent using stoc. gradients. It is not clear whether our proof directly extends to the nonconvex-concave setting. The only convergence analysis we know for sm-AGDA in the nonconvex-concave setting is given in [68]; but, this analysis is given only for the deterministic case and requires the compactness of the dual domain. Yet, we believe that combining our analysis techniques with some deterministic inequalities from [68] may allow to extend our results to the nonconvex-concave setting. We will consider this as part of future work as it would require a detailed analysis with some fundamental differences from the current work.
>
> **3-** In stoc. GDA, the primal and dual updates can be implemented either in an alternating fashion or in a Jacobi style where both updates are performed simultaneously. The alternating case is referred to as stoc. alternating GDA (AGDA). By slightly modifying our proof, we can show a similar HP result holds for stoc. AGDA, with a degraded complexity of $O(\ell \kappa^2 / \epsilon^2 + \delta^2 \log(1/q) \kappa^2 / \epsilon^2 + \ell \kappa^4 \delta^2 / \epsilon^4)$. Our stoc. AGDA’s complexity has extra $\kappa$ factors on the bias and variance/quantile error parts, respectively, when compared to sm-AGDA bound we derive in our paper. To obtain the stoc. AGDA bound, we apply Thm. 9 to the modified stochastic processes: $\tilde A_t = \tau_1 V_t, \tilde B_t = \frac{\tau_1}{32} \|\| \nabla \Phi(x_t) \|\|^2 + \frac{\tau_2}{64} \|\| \nabla f(x_t, y_t)\|\|, \tilde D_t = \kappa \ell\tau_1^2\|\| \Delta_t^x \|\|^2 + \ell\tau_2^2 \|\| \Delta_t^y \|\|^2/8$,
> $$\tilde C_t = \left\langle \left( (1 + \alpha) \frac{2\ell + \ell \kappa}{2} \tau_1^2 - \alpha \left(\tau_1 + l \tau_1^2 - \tau_2 \ell^2 \tau_1^2 \right) \nabla_x f(x_t, y_t) - (1 + \alpha) \tau_1 \nabla \Phi(x_t) \right), \Delta_t^x \right\rangle - \left\langle \alpha (\tau_2 - \ell \tau_2^2) \nabla_y f(x_t, y_t), \Delta_t^y \right\rangle,
> $$
> where $V_t = (1 + \alpha) \Phi(x_t) - \alpha f(x_t, y_t)$ is a Lyapunov function, $\alpha$ is a particular constant. Jacobi-style stoc. GDA is known to be divergent even in the convex-concave setting [80], so we did not consider it. We also added the missing references.
>
> **References**
>
> [80] Zhang, Wang, Lessard, Grosse. AISTATS 2022.
>
> [81] Harvey, Liaw, Plan, Randhava, COLT 2019.
>
> [82] Rakhlin, Shamir, Sridharan, ICML 2012.

---

> > ### Comment · Reviewer_pQzv · 2024-08-13
> >
> > Thanks for the response. I will keep my score.

---

### Official Review · Reviewer_vFP9 · 2024-07-12

**Soundness:** 2
**Presentation:** 2
**Contribution:** 2
**Rating:** 4
**Confidence:** 4

**Summary:**

This paper investigates the stochastic non-convex minimax optimization problem $\min_x \max_y f(x,y)$ where the function $f$ is smooth, then nonconvex in $x$ and PL in $y$. The authors presented a high probability analysis for the smoothed alternating gradient descent ascent (sm-AGDA) method. This analysis is based on a Lyapunov function from prior work for non-convex concave setting. Also, under a light-tail assumption on the gradient noise, the authors developed a concentration result. Finally, they include experiments of nonconvex-PL setting with synthetic data and then distributionally robust optimization problems with real data.

**Strengths:**

The authors provided theoretical proofs and experiments to support their claims. The analysis seems to be solid. The authors compared their complexity with other high probability bounds results.

**Weaknesses:**

1. While the authors cited some results in expectation for the similar setting, I think it is important to compare the (Markov) high probability bounds obtained by these expectation results with the results obtained in this paper. If the results are not better, then the contribution is limited.

Also I don't get what the authors meant by "More precisely, we focus on a purely stochastic regime in which data streams over time which renders the use of mini-batch schemes or running the method in parallel impractical; therefore, approaches based on Markov’s inequality [59] are no longer applicable." (line 99-101).

2. In the theorems and lemma, the values $\tau_1, \tau_2, \tau, \bar{\tau}$ is really confusing. Do you need to pick specific values for the step sizes, if so, what is $\tau_2$? Are $\tau$ and $\bar{\tau}, \tau_1$ the same?

Update: I thank the authors for the rebuttal. I keep my recommendation the same, as changing to high probability bound from expectation results is somewhat expected.

**Questions:**

Why in the experiments the authors compare with SAPD+ and SMDAVR, not the methods in Table 1?

---

> ### Author Rebuttal · Authors · 2024-08-04
>
> We thank the referee for their time invested in reviewing our paper. Below are our responses to the questions raised.
>
> **Weaknesses**
>
> **1-** Please see our general response to the reviewers above in the window titled **General response: Tightness of our high-probability bounds,** where we explain why our bounds are significantly tighter than any naive bound based on applying Markov's inequality to the existing results in expectation. As we discuss in detail there, applying Markov's inequality yields significantly loose bounds in terms of their dependence on the quantile parameter $q \in (0,1)$.
>
> A more advanced approach - see e.g. [19,59] - involves running $O(\log(1/q))$ copies of the sm-AGDA algorithm together with a postprocessing step requiring additional $\mathcal{O}(1/\epsilon)$ samples for each copy to identify an $\epsilon$-stationary point with high probability.
> The overall cost of this approach is of order $O(\log(1/q)) \cdot \mathcal G(\epsilon) + O(\log(1/q)) \cdot O(1/\epsilon) $, where $\mathcal G(\epsilon)$ denotes the complexity in expectation, assuming stochastic gradients have a variance bounded by $\sigma^2$. This results in an unavoidable dependence in the failure probability $q$ of order $\log(1/q) \epsilon^{-4}$ which is significantly worse than our $\log(1/q) \epsilon^{-2}$ dependance.
>
> Moreover, approaches like [19,59] can remain interesting in practice when one is allowed to have the $O(\log(1/q))$ runs of sm-AGDA processed in parallel, but loses all interest with respect to our approach when parallelization is impractical or impossible.  Indeed, the approaches that require $O(\log(1/q))$ parallel runs can be impractical for the streaming data setting we consider where the data arrives one by one in a streaming fashion; the aforementioned impracticality is mainly because of the fact that the parallel runs would have to wait for the arrival of  new data points to be able take one step. This is what we meant to convey in our sentence "*More precisely, we focus on a purely stochastic regime in which data streams over time, rendering the use of mini-batch schemes or running the method in parallel impractical, therefore ... [59] are no longer applicable*". In the revised version, we clarified this sentence and added comparisons with naive approaches based on the Markov inequality.
>
> **2-** While $\tau_1$ and $\tau_2$ are primal and dual stepsizes, $\bar{\tau}$ corresponds to a particular choice of the primal stepsize where we set $\tau_1=\bar{\tau}$ in our main complexity result. To avoid confusion, we will remove $\bar{\tau}$ and use only $\tau_1$ instead. The symbol $\tau$ is used in our concentration inequality, to emphasize that it can in principle apply to any positive scalar (not necessarily the primal stepsize); however in practice we choose $\tau=\tau_1$. Therefore, to simplify the notation we will also replace $\tau$ with $\tau_1$ in the revised version.
>
> **Questions**
>
> **1-** The methods provided in the table are mostly for convex/concave problems and do not provide any guarantees for the stochastic nonconvex/strongly convex (NCSC) problem we consider in this experiment. There are also some VI results in the table, but they do not apply to NCSC problems either. The point of the table was to summarize the existing HP results and emphasize that no HP bounds were known for NCSC problems prior to our paper. SAPD+ and SMDAVR algorithms have in expectation guarantees for NCSC problems, but they do not admit HP guarantees. SAPD+ and SMDAVR were also among the most competitive ones tested on this example in [70]. This is the main reason why we compared sm-AGDA with these two algorithms even if they do not admit HP guarantees.
>
> We hope these responses satisfy the reviewer and lead to an improvement in our score.

---

### Official Review · Reviewer_Zwn2 · 2024-07-12

**Soundness:** 3
**Presentation:** 3
**Contribution:** 2
**Rating:** 5
**Confidence:** 3

**Summary:**

The authors build upon earlier works to prove a high probability upper bound, that is linear in variance of the stochastic gradients, on the number of stochastic gradient calls of smoothed alternating GDA method.

**Strengths:**

The paper is well written with clear theorems, proofs and numerical results.

**Weaknesses:**

Focusing only on their concentration bound, which they claim is of independent interest and is the crux of their high probability bounds: the bound in Theorem 9 (which is fairly customized and not easy to state) follows like any other chernoff/chebychev type bound - using Markov type inequality after exponentiation; and I don't see any novelty in their theoretical ideas/techniques here. Perhaps I have missed something, if so the authors can clarify it.

**Questions:**

see weaknesses

**Limitations:**

Theory paper, no concerns

---

> ### Author Rebuttal · Authors · 2024-08-05
>
> We thank the referee for their time invested in our paper. Below are our responses to the weaknesses raised.
>
> While our concentration inequality (Thm 9) seems tailored to the analysis of sm-AGDA, we can argue that it can also aid in deriving high probability bounds for many other nonconvex first-order methods that outputs a randomized iterate. Indeed, most Lyapunov arguments in the nonconvex setting are built upon telescoping quantities in line with our Thm 9. This includes stochastic alternating GDA for NCPL minimax problems and optimistic GDA [34] for strongly convex-strongly concave problems. The challenge lies in finding the appropriate stochastic processes $(\tilde{A}_t, \tilde{B}_t, \tilde{C}_t, \tilde{D}_t)$, as done in Cor 8 for sm-AGDA. This is akin to prime factorization—easy to verify but hard to come up with. Thus, our significant technical novelty is devising these processes for sm-AGDA such that the concentration result in Thm 9 applies. We mentioned Thm 9 to be of independent interest, not because it is particularly hard to prove but because it acts as a useful inequality potentially applicable to other algorithms like stochastic AGDA and stochastic gradient descent (SGD).
>
> For example, stochastic AGDA is a special case of the sm-AGDA algorithm we analyzed, obtained by setting the parameters $\beta=p=0$. Thm 9 leads to a new complexity result of  $O(\ell \kappa^2 / \epsilon^2 + \delta^2 \log(1/q) \kappa^2 / \epsilon^2 + \ell \kappa^4 \sigma^2 / \epsilon^4)$ for stochastic AGDA, if we were to apply Thm 9 to processes $\tilde A_t = \tau_1 V_t, \tilde B_t = \frac{\tau_1}{32} \|\| \nabla \Phi(x_t) \|\|^2 + \frac{\tau_2}{64} \|\| \nabla f(x_t, y_t)\|\|, \tilde D_t = \kappa \ell\tau_1^2\|\| \Delta_t^x \|\|^2 + \ell\tau_2^2 \|\| \Delta_t^y \|\|^2/8$ together with an appropriately modified Lyapunov function $V_t$ and $\tilde C_t$ term. Similarly, by adapting these processes and by removing the dual terms that relate to $y$ variable, we can obtain complexity results for SGD.
>
> Thanks to the novelty of our approach, we can also obtain tight bounds as we discuss next. Specifically, our complexity bound for sm-AGDA for finding an $\epsilon$-stationary solution w.p. at least $1-q$ is $O \left(\ell \kappa/\epsilon^2 + \delta^2 \log\left(1/q\right) \kappa/\epsilon^2 + \delta^2 \ell \kappa^2/\epsilon^4\right),$ where $\kappa = \ell/\mu$ is the condition number, and $\delta^2$ is the subGaussian proxy tied to the stochastic gradient estimates $\widetilde{\nabla}_x f, \widetilde{\nabla}_y f$. These bounds are tight in the sense that when $\delta=0$, we recover the deterministic $O(\kappa/\epsilon^2)$ complexity for sm-AGDA; and the quantile term scales favorably with $\epsilon$ and $q$. One may expect the iterates to have light-tail properties when the gradient noise is light-tailed, but quantifying the proxy/variance of the iterates is not straightforward since it depends on various algorithm and problem parameters such as $\mu,\ell,\tau_1,\tau_2,\beta$, etc. Furthermore, turning in-expectation estimates to high-probability (HP) estimates via Markov inequality would result in much worse estimates than ours, as we explain in the window titled **General response: Tightness of our high-probability bounds**. As we discuss in this window, a surprising result of our analysis, contrary to advanced approaches leveraging Markov's inequality by running the method several times in parallel, is that the cost of getting an HP bound only affects negligible terms of the initial expectation bounds giving $O(\epsilon^{-4}+\log(1/q)\epsilon^{-2})$ complexity, instead of multiplying the whole in-expectation bound by $\log(1/q)$ in which case one gets a much worse $O(\log(1/q)\epsilon^{-4})$ complexity. This feature departs from standard guarantees developed for the last iterate in the SCSC setting [79,34].
>
> Our bounds achieve good complexity by striking the right balance between the bias and variance/quantile terms in the complexity. This is accomplished by carefully defining the stochastic processes $\tilde{A}_t, \tilde{B}_t, \tilde{C}_t, \tilde{D}_t$ in our Cor 8 that exhibit the desired concentration properties as stated in our generic concentration result (Thm 9). In particular, these processes depend on the parameters $\tau_1, \tau_2,$ and $\beta$ in non-trivial ways, requiring parameter choice/design optimization to achieve our rate results. Naive choices would result in a much worse complexity bound in terms of its dependence on $\epsilon, \kappa, $ and $q$. In addition, constructing the processes $\tilde{A}_t, \tilde{B}_t, \tilde{C}_t, \tilde{D}_t$ requires careful measurability considerations along with deriving various inequalities (that carefully exploit the smoothness and PL properties of the objective) to extend the existing analysis for deriving expectation guarantees [63] and for the deterministic setting [68] of sm-AGDA. For a discussion on the challenges related to measurability in the Lyapunov analysis of momentum-averaged algorithms, we refer to [34], which obtained HP bounds in the strongly convex-strongly concave (SCSC) setting. Similarly, sm-AGDA employs momentum averaging using the parameter $\beta$ in the nonconvex setting and encounters comparable measurability issues. These challenges are specific to HP analyses and do not occur in the less sensitive expectation bounds from [64], where noise can be mitigated using the properties of conditional expectation. Therefore, substantial amount of new work is in fact needed to obtain our results.
>
> To summarize, our bounds are highly non-trivial requiring significant novelty, where we get the state-of-the-art complexity for nonconvex/PL minimax problems by far. We would appreciate if the referee would consider raising their score.
>
> **References:**
>
> [79] Cutler, Drusvyatskiy, and Harchaoui. Stochastic optimization under time drift: iterate averaging, step-decay schedules, and high probability guarantees. NeurIPS, 2021.

---

> > ### Comment · Reviewer_Zwn2 · 2024-08-13
> >
> > > We mentioned Thm 9 to be of independent interest, not because it is particularly hard to prove but because it acts as a useful inequality potentially applicable to other algorithms like stochastic AGDA and stochastic gradient descent (SGD).
> >
> > I have increased the score, but please clarify the above in the main body of the paper, so that your contributions are clear.

---

### Official Review · Reviewer_H241 · 2024-07-12

**Soundness:** 4
**Presentation:** 3
**Contribution:** 3
**Rating:** 7
**Confidence:** 5

**Summary:**

This paper considers the important open problem of stochastic smooth nonconvex minimax optimization. This paper proposes single-loop stochastic GDA method, which was known to be practically desirable but had no theoretical complexity compared to other non-single-loop methods with better complexies on nonconvex minimax problems. The analysis in this work fills some of above gap, provides the first high-probability complexity for nonconvex minimax while assuming a PL condition on dual variable, and proves assuming light tailed stochastic gradients GDA converges to a near stationary point with a certain complexity. Numerical results on NCPL game with synthetic data and distributionally-robust optimization with real data shows the proposed sm-AGDA outperforms existing algorithms SAPD+ and SMDAVR.

**Strengths:**

1. This paper solves the important open problem of stochastic smooth nonconvex minimax optimization, proposes a single-loop GDA method called sm-AGDA, which is constructed by Lyapunov function for nonconvex-concave problems.

2. Assuming PL condition, this paper proved the first high-probability complexity bound on such single-loop algorithm on nonconvex minimax problem. The order of complexity is reasonably good.

3. Numerical experiments show the proposed method is practically superior to existing methods.

**Weaknesses:**

1. Font size at the end of Page 8, and entire Page 9 are smaller than the required size.

2. There is not much algorithm novelty from [68].

**Questions:**

1. What are the existing best-known complexities of (possibly non-single-loop) algorithms on nonconvex minimax problem?

2. Can you compare in more details (overall problem class, overall algorithm, assumptions, complexity order dependencies) of your complexity result compared with [64]?

3. Does each of your assumption (PL condition, etc.) hold for both of your numerical experiments?

**Updates:

The authors' rebuttal addressed every question very well. I have increased my score from 6 to 7.

---

> ### Author Rebuttal · Authors · 2024-08-05
>
> We thank the referee for their time invested in reviewing our paper. Below are our responses to the questions raised.
>
> **Weaknesses**:
>
> **1-** The smaller font size is a typo due to misplaced bracket after a displayed equation. We will fix this issue; thanks for the good catch.
>
> **2-** Our work extends sm-AGDA from [68] by allowing light-tail stochastic estimates $\widetilde{\nabla}_x f, \widetilde{\nabla}_y f$ in place of the exact partial gradients. This is relevant for large-scale optimization and machine learning where gradients are often estimated from streaming or random data samples. sm-AGDA's convergence in the stochastic setting was first established in expectation in [64]. Our work provides its **first** high-probability (HP) bounds for stochastic nonconvex-PL problems.
>
> A substantial body of work has been dedicated to establishing HP guarantees for first-order methods. These works are often theoretical and involve complex arguments to achieve tight bounds, even in the minimization setting for methods like stochastic gradient descent (SGD) [77,78]. Our paper falls within this line of research.
>
> Our main result (Thm 10) is a tight sampling complexity bound for finding an $\epsilon$-stationary solution with probability at least $1-q$ on nonconvex-PL problems. Our bounds are tight as when $\delta=0$ (i.e. gradients estimates are exact), we recover the deterministic complexity of sm-AGDA [68]; and the quantile part of our bound (involving $\delta^2$) scales favorably with $\epsilon$ and $\kappa$. Turning expectation estimates to HP estimates via standard Markov approaches would result in much worse estimates than ours, as we explain in our **General response: Tightness of our high-probability bounds**.
>
> Our key technical contribution lies in identifying stochastic processes ($\tilde{A}_t, \tilde{B}_t, \tilde{C}_t, \tilde{D}_t$ in Cor 8) with desired concentration properties (Thm 9). Constructing these processes (e.g., $\tilde{C}_t$) is non-trivial, requiring careful measurability and deriving various inequalities (exploiting smoothness and PL properties) to extend the existing analysis for deriving expectation guarantees [63] and for the deterministic setting [68] of sm-AGDA. For a discussion on related measurability challenges in Lyapunov analysis of momentum-averaged methods, see [34] which obtained HP bounds in the SCSC setting.
>
> **Questions**
>
> **1-** In this paper, we considered stochastic smooth nonconvex-PL (NCPL) problems. For deterministic NCPL problems, AGDA and sm-AGDA have the complexity of $O(\kappa^2/\epsilon^2)$ and $O(\kappa/\epsilon^2)$ respectively for finding a point $(x,y)$ satisfying $\\|\nabla f(x,y)\\|\leq \epsilon$ as shown in [68]. Here $\kappa = L/\mu$ is the condition number, where $\ell$ is the Lipschitz constant of the gradient, and $\mu$ is the PL constant. For Catalyst-AGDA, [64] shows also the rate $O(\kappa/\epsilon^2)$ for deterministic NCPL problems. Regarding stochastic NCPL problems, the only existing result holds in expectation by [64]; the authors show that stochastic AGDA and stochastic sm-AGDA have the complexity of $O(\kappa^4/\epsilon^4)$ and $O(\kappa^2/\epsilon^4)$ respectively for computing a point $(x,y)$ which satisfies $\mathbb{E}\\|\nabla f(x,y)\\|\leq \epsilon$.
>
> Our paper provides the first HP results for stochastic NCPL problems, showing that the stochastic sm-AGDA method can compute a point $(x,y)$ that satisfies $\|\|\nabla f(x,y)\|\| \leq \epsilon$ w.p. at least $1-q$ within $O\left(\ell\kappa^2\delta^2 \epsilon^{-4} + \kappa \epsilon^{-2} (\ell + \delta^2 \log(1/q))\right)$ stochastic gradient calls for any $q \in (0, 1)$.
>
> **2-** In our paper, we consider the same algorithm (sm-AGDA) under the same assumptions (smooth NCPL problems) considered in [64]. Assuming that the variance of the stochastic gradient is bounded by a constant $\delta^2$, [64] shows a complexity of  $O\left(\ell\kappa^2\delta^2 \epsilon^{-4} + \kappa \epsilon^{-2} \ell \right)$ stochastic gradient calls for computing $(x,y)$ that satisfies $\mathbb{E}\\|\nabla f(x,y)\\|\leq \epsilon$. In our work, we make an additional light-tail (subGaussian) assumption on the noise, and obtain a HP result showing that $O\left(\ell\kappa^2\delta^2\epsilon^{-4} + \kappa \epsilon^{-2} (\ell + \delta^2 \log(1/q))\right)$ stochastic gradient calls are sufficient to calculate a point $(x,y)$ satisfying $\|\|\nabla f(x,y)\|\| \leq \epsilon$ w.p. at least $1-q$. Such light-tail assumptions are commonly made for HP results even in convex minimax problems as we discuss in the introduction.
>
> To our knowledge, this is the first HP result for a nonconvex minimax problem as we highlight in the introduction/Table 1. In particular, one should note that HP guarantees provide a much finer resolution than in-expectation ones, since gradients can be small on average (in expectation), while still being arbitrarily large with some positive probability. Finally, our HP bounds prove to be tight as we recover the in expectation one from [64] using the fact that the expectation of any random variable can be written as the integral of its quantile function from $p=0$ to $p=1$ : $\mathbb{E}[U] = \int_{p=0}^1 Q_p(U) dp$.
>
> **3-** Yes. The NCPL assumption holds for the first experiment directly; here the primal problem is indeed non-convex and the dual is PL (and not strongly concave). The second experiment is in the non-convex/strongly concave (NCSC) setting as the regularizer $g(y)$ is strongly convex. Since SC implies the PL property; the second experiment in the NCSC setting is a special case of the NCPL setting. As such, our assumption of NCPL objectives hold for both of our experiments.
>
> We hope these responses satisfy the reviewer and result in an improved score.
>
> **References**
>
> [77] A.R., O.S., K.S. Making gradient descent optimal for strongly convex stochastic optimization. ICML 2012.
>
> [78] N. H., et al. "Tight analyses for non-smooth stochastic gradient descent." COLT 2019.

---

> > ### Comment · Reviewer_H241 · 2024-08-08
> > **Response to authors**
> >
> > I believe the authors addressed all 5 of my questions very well. I hope the authors will add those valuable discussions in the appendix. Based on the rebuttal, I update my score from 6 to 7.

---

### Official Review · Reviewer_NU5h · 2024-07-14

**Soundness:** 3
**Presentation:** 3
**Contribution:** 2
**Rating:** 5
**Confidence:** 3

**Summary:**

The authors derive a high probability bound for convergence of a stochastic gradient descent-ascent method over a nonconvex (PL class) of functions. Specifically, they analyze the sm-AGDA algorithm, which previously only had a bound in expectation, and show a similarly tight high probability bound when gradient noise is assumed subgaussian. The authors provide two experimental settings demonstrating that the distribution of these iterates over multiple runs converge as expected for the algorithm.

**Strengths:**

* Technically solid paper that thoroughly analyzes the sm-AGDA algorithm and understandably interprets the results (e.g. remark 11). I don't think the assumptions are too strict, or at least not outside the norm for this type of analysis.

* Good comparison to other related works, especially Table 1 is useful to gauge the current state of theoretical results for this problem.

* Helps to move analysis closer to realistic nonconvex min-max optimization settings.

**Weaknesses:**

* While I understand that the bound in Thm 10 over the average gradient norms motivates the random sampling of iterates in the sm-AGDA algorithm, it still seems like practically the best thing to do in Fig. 1 is to take the last iterate (or at least sample over some last window of iterates). Is there a straightforward way to come up with a similar high probability bound on the last iterate?

* I'd maybe like to see more discussion about how the analysis in this paper differs from that in the original sm-AGDA work [63]. At least for some previous work I've seen with high probability bounds for convex optimization, oftentimes the actual modification required to go from expectation to high probability is fairly minor (i.e. just boils down to bounding the summation of Subgaussian noise terms).

* I like the first set of experiments but a bit confused at the purpose of the second. I don't quite understand comparing concentration properties of other methods if the analysis only applies to one algorithm. Maybe if the authors highlighted what unique components of the sm-AGDA algorithm might give it better quantile convergence properties than others this would make more sense.

If the authors can address some of these concerns I am willing to raise my score to a more solid accept

Other Minor Notes:
* DRO abbreviation in line 238 not defined initially
* [63] and [64] appear to be referencing the same paper
* Line 306 and 311 appear to be referencing the wrong figure

**Questions:**

* As mentioned in remark 11 this bound appears to be pretty tight compared to the bound in expectation. Do you think if this type of analysis was applied to other existing algorithms you would get similarly tight bounds?

**Limitations:**

The authors clearly state their assumptions.

---

> ### Author Rebuttal · Authors · 2024-08-05
>
> We thank the reviewer for all the feedback. Below we provide a point-by-point response to each of the weakness/question raised in order.
>
> **Weaknesses**
>
> **1-** The only known last-iterate results for nonmonotone VI problems require more restrictive conditions like local *quadratic growth* around critical points or 2nd-order sufficient conditions [28, 74]. Some existing worst-case analysis in convex-concave cases shows that complexity with averaging can be strictly better than the last iterate [76]. In conclusion, establishing a last-iterate convergence result in our nonconvex setting appears to be difficult and unlikely to offer better guarantees than our averaging approach.
>
> **2-** We concur with the reviewer that noise from subGaussian gradient estimates are likely to produce a light-tailed distribution for the final iterate (or its average), but the main challenge remains achieving a precise balance within the bias-quantile trade-off of our complexity. Our bounds are tight as when $\delta=0$, we recover the deterministic $O(\kappa/\epsilon^2)$ bound for sm-AGDA; and the quantile part (i.e. terms involving $\delta$) of our bounds scales favorably with $\epsilon$ and $q$. Turning expectation estimates to high-probability (HP) estimates via Markov inequality would result in much worse estimates than ours, see our **General response: Tightness of our high-probability bounds**.
>
> Our bounds achieve good complexity by striking the right balance between their bias and quantile terms. This is done by defining stochastic processes ($\tilde{A}_t, \tilde{B}_t, \tilde{C}_t$, and $\tilde{D}_t$ in Cor. 8) that show the desired concentration properties (Thm. 9). Constructing these processes (e.g., $\tilde{C}_t$) is non-trivial, requiring careful measurability considerations and deriving various inequalities (exploiting smoothness and PL properties) to extend the existing analysis for deriving expectation guarantees [63] and for the deterministic setting [68] of sm-AGDA. For a discussion on measurability challenges in the HP analysis of momentum-averaged algorithms, see [34] which obtained HP bounds in the SCSC setting. Similarly, sm-AGDA employs momentum averaging through its parameter $\beta$ in the nonconvex setting and faces similar measurability issues. These challenges are specific to HP analyses and do not occur in the less sensitive expectation bounds from [63], where noise can be mitigated using the properties of conditional expectation.
>
> Obtaining HP guarantees in the nonconvex min-max problems is significantly harder than for nonconvex unconstrained minimization due to the necessary time-scale separation between the primal and dual, where primal updates leading to a descent in the primal function can amplify errors in the dual domain if parameters and descent properties are not carefully designed/analyzed [75,6,39].
>
> **3-** In our experiments, both SAPD+ and SMDAVR needed smaller primal and dual step sizes and more tuning of them. Generally, for GDA methods, $\tau_1$ and $\tau_2$ must be chosen such that $\tau_1/\tau_2=\Omega(\kappa)$. For stoc. GDA [39] and stoc. AGDA [6], $\tau_1/\tau_2=\Theta(\kappa^2)$, but sm-AGDA does not require this thanks to its primal regularization which further allows working with larger primal and dual step sizes of the same order. For example, in sm-AGDA, one can use a primal step size $\tau_1 = O\left(\min\left(\frac{1}{\ell}, \frac{1}{\sqrt{T}}\right)\right)$ and a dual step size $\tau_2=\Theta(\tau_1)$. These factors contribute to sm-AGDA's practical success and to why it can admit better quantile convergence properties.
>
> Reviewer is correct that our quantile bound analysis only applies to sm-AGDA. However, we compare sm-AGDA's concentration properties with SAPD+ and SMDAVR to provide a baseline for sm-AGDA results. SAPD+ and SMDAVR have state-of-the-art complexity bounds in expectation (SAPD+ [70] has better condition number dependency than sm-AGDA [63]). Yet, their quantile bounds have not been studied. We wanted thus to see sm-AGDA's practical performance against these methods.
>
> sm-AGDA is a state-of-the-art method among single-loop algorithms for nonconvex-PL minmax problems in both deterministic [68] and in-expectation performance metrics [63]. For example, sm-AGDA offers better in-expectation guarantees than stochastic (alternating) GDA in terms of the complexity bound's dependence on the condition number $\kappa$. Single-loop methods like sm-AGDA are favored for their simplicity when compared to multi-loop algorithms, e.g., [70]; hence, they are easy to tune. These factors contribute to why sm-AGDA may achieve better quantile properties than other methods like stoc. GDA or multiple-loop algorithms.
>
> **4-** We addressed the minor notes, thanks for the good catch.
>
> **Questions**
>
> **1-** Indeed, our approach can adapt to other algorithms for nonconvex-PL problems once one defines new stochastic processes $\tilde A_t, \tilde B_t, \tilde C_t, \tilde D_t$ (as in Cor. 8) and adjusts the potential function. For example, our analysis applies to stoc. AGDA, but results in worse HP bounds compared to sm-AGDA in terms of their dependence to the condition number $\kappa$, i.e., $\kappa$ factor in the sm-AGDA complexity becomes $\kappa^2$ for stoc. AGDA.
>
> We hope these responses satisfy the reviewer and lead to an improvement in our score in terms of a solid accept.
>
> ## References:
> [74] W. Azizian, F. Iutzeler, J. Malick, and P. Mertikopoulos. The rate of convergence of bregman proximal methods: Local geometry versus regularity versus sharpness. SIAM Journal on Optimization, 34(3):2440–2471, 2024.
>
> [75] H. Li, F. Farnia, S. Das, and A. Jadbabaie. On convergence of gradient descent ascent: A tight local analysis. ICML, pages 12717–12740. 2022.
>
> [76] N. Golowich, S. Pattathil, C. Daskalakis, and A. Ozdaglar. Last iterate is slower than averaged iterate in smooth convex-concave saddle point problems. COLT, pp. 1758–1784. 2020.

---

### Author Rebuttal · Authors · 2024-08-04

### General response: Tightness of our high-probability bounds

We thank the referees for their time invested in reviewing our paper. Following the request from several referees, we discuss below the benefit of our approach against naive expectation-to-high-probability conversions using Markov's inequality.

Our bounds are significantly better than the simple high-probability bounds one can obtain using the existing results in terms of expectation; this point is explained in detail below. For the nonconvex-PL setting considered in our paper, the only expectation result we are aware of is by [64]. Assuming the variance of the stochastic gradient is bounded by $\delta^2$, for sm-AGDA the authors show a complexity of $\mathcal{G}(\epsilon):=O\left(\frac{\ell\kappa^2\delta^2}{\epsilon^4} + \frac{\kappa}{\epsilon^2}\ell \right)$ iterations/stochastic samples for computing an $\epsilon$-stationary solution in expectation, i.e., for computing $(x,y)$ that satisfies $\mathbb{E}\\|\nabla f(x,y)\\|\leq \epsilon$. Here, $\mu$ is the PL constant, $\ell$ is the gradient Lipschitz constant and $\kappa=\ell/\mu$. This result basically acts as an oracle that generates a sample $(\hat x,\hat y)$ satisfying $\mathbb{E}\\|\nabla f(\hat x,\hat y)\\| \leq \epsilon$ after $\mathcal{G}(\epsilon)$ iterations. We next present some naive approaches using the Markov inequality to obtain trivial high-probability bounds that hold w.p. at least $1-q$ for any given $q\in (0,1)$:

- The first approach involves applying the oracle to generate a solution $(\hat x,\hat y)$ such that $\mathbb{E}[\\|\nabla f(\hat{x}, \hat{y})\\|] \leq q \cdot \epsilon$. Then by naively using Markov's inequality, we get an $\epsilon$-stationary point w.p. at least $1-q$. The complexity bound for this approach is $\mathcal{G}(q\epsilon) = O\left(\frac{\ell\kappa^2\delta^2}{q^4\epsilon^4} + \frac{\kappa}{\epsilon^2q^2}\ell \right)$ and it scales badly with $q$.
- The second approach involves calling the oracle $m$ times to generate a group of $\epsilon/2$-stationary solutions $\\{(\hat x^{(i)}, \hat y^{(i)})\\}_{i=1}^m$. If the performance metric $\\| \nabla f (\hat x^{(i)}, \hat y^{(i)})\\|$ can be evaluated to $\epsilon$-accuracy for all $i\in\{1,\ldots,m\}$, i.e., for each $i$, we get a stochastic estimate $\tilde\nabla{f (\hat x^{(i)}, \hat y^{(i)})}$ such that $\\|\tilde\nabla{f (\hat x^{(i)}, \hat y^{(i)})}-\nabla{f (\hat x^{(i)}, \hat y^{(i)})}\\| \leq \epsilon$ with high probability, then the point $(\hat x^{(i^*)}, \hat y^{(i^*)})$ such that $i^* = \arg\min\\{\\|\tilde\nabla{f (\hat x^{(i)}, \hat y^{(i)})}\\| : i=1,\ldots, m\\}$ will satisfy the desired high-probability bound as long as $m = \Omega \left( \log \frac{1}{q} \right)$. This approach resolves the unfavorable dependence on $q$ present in the first approach. However, evaluating the stochastic estimate $\tilde\nabla{f (\hat x^{(i)}, \hat y^{(i)})}$ with the aforementioned properties for each $i$ typically requires $O(1/\epsilon^2)$ samples if light-tail assumptions are not made for the stochastic gradients [36, 59], [Thm 2.4,19] (using Markov inequality-type bounds or standard concentration inequalities such as [Lem 2.3,19]); on the other hand, the number of samples required for each $i$ is $O(\delta^2/\epsilon)$ if light-tail (subGaussian) assumptions are made for the stochastic gradients ignoring some logarithmic factors [Cor. 2.5,19]. In our paper, we consider general problems *without* assuming a finite-sum form for the objective and the data can be arriving in a streaming fashion. Therefore, the total complexity with the second approach when applied to the problems we consider becomes at least $\log\left(\frac{1}{q}\right) \mathcal{G}(\epsilon) + \log\left(\frac{1}{q}\right) O\left(\frac{\delta^2}{\epsilon}\right) = O\left(\log\left(\frac{1}{q}\right)\frac{\ell\kappa^2\delta^2}{\epsilon^4} + \log\left(\frac{1}{q}\right)\frac{\kappa\ell}{\epsilon^2} + \log\left(\frac{1}{q}\right)\frac{\delta^2}{\epsilon}\right)$.
- For unconstrained strongly convex-strongly concave (SCSC) problems, one can use the robust distance estimation technique together with also $m = \Omega \left( \log \frac{1}{q} \right)$ parallel runs; however, this approach does not apply to nonconvex minimax problems we consider in this paper [Sec 2,36].

Our high-probability result shows a complexity of $O\left(\frac{\ell\kappa^2\delta^2}{\epsilon^4} + \frac{\kappa}{\epsilon^2}\Big(\ell + \delta^2 \log(\frac{1}{q})\Big)\right)$ under the light-tail (subGaussian) assumption. This result is significantly better than the approaches mentioned above; indeed, it has not only the best (logarithmic) scaling with respect to $q$ but also with respect to $\epsilon$. In the second approach, the logarithmic term $\log(1/q)$ multiplies the high-order $O(1/\epsilon^4)$ term, whereas in our approach it only affects the second-order $O(1/\epsilon^2)$ term. More importantly, the approaches that require $\log(1/q)$ parallel runs can be impractical for the streaming data setting we consider where the data arrives one by one in a streaming fashion; the aforementioned impracticality is mainly because of the fact that the parallel runs would have to wait for the arrival of $\Omega(\log(1/q))$ new data points to be able take one step.

---

### Decision · Program_Chairs · 2024-09-25

**Decision:**

Accept (poster)

**Comment:**

This paper addresses the high probability convergence of the Gradient Descent-Ascent-type method for stochastic nonconvex minimax optimization. The problem studied is PL in the max variable and hence the inner problem is computationally "easy" to solve. Having easy inner problem and working with the dual function makes the problem easier than some other settings to analyze (in min-max optimization). By deriving high probability bounds, the paper extend previous results that only provided bounds in expectation. The analysis also introduces a new concentration bound  that might be of independent interest.  The paper includes numerical experiments comparing their studied algorithms with existing methods.

Some reviewers found the theoretical techniques (particularly the concentration bounds) not significantly novel, as it follows existing bounds/techniques. Some reviewers believe that the paper could benefit from a deeper comparison with related works, especially regarding the differences between the current and prior analyses.  There were also some concern that the algorithmic contributions have limited insight into new theoretical developments beyond those already present in the literature.

Overall, the paper is borderline. While the paper makes contributions to the theoretical understanding of GDA-type algorithms for min-max optimization, addressing the above concerns could strengthen its impact.